# Normative decision rules in changing environments

**Nicholas W Barendregt[1]\*, Joshua I Gold[2], Krešimir Josić[3], Zachary P Kilpatrick[1]**

[1]Department of Applied Mathematics, University of Colorado Boulder, Boulder, United States; [2]Department of Neuroscience, University of Pennsylvania, Philadelphia, United States; [3]Department of Mathematics, University of Houston, Houston, United States

**Abstract** Models based on normative principles have played a major role in our understanding of how the brain forms decisions. However, these models have typically been derived for simple, stable conditions, and their relevance to decisions formed under more naturalistic, dynamic conditions is unclear. We previously derived a normative decision model in which evidence accumulation is adapted to fluctuations in the evidence-generating process that occur during a single decision (Glaze et al., 2015), but the evolution of commitment rules (e.g. thresholds on the accumulated evidence) under dynamic conditions is not fully understood. Here, we derive a normative model for decisions based on changing contexts, which we define as changes in evidence quality or reward, over the course of a single decision. In these cases, performance (reward rate) is maximized using decision thresholds that respond to and even anticipate these changes, in contrast to the static thresholds used in many decision models. We show that these adaptive thresholds exhibit several distinct temporal motifs that depend on the specific predicted and experienced context changes and that adaptive models perform robustly even when implemented imperfectly (noisily). We further show that decision models with adaptive thresholds outperform those with constant or urgency-gated thresholds in accounting for human response times on a task with time-varying evidence quality and average reward. These results further link normative and neural decision-making while expanding our view of both as dynamic, adaptive processes that update and use expectations to govern both deliberation and commitment.

**\*For correspondence:**
nicholas.barendregt@colorado.edu

## Editor's evaluation

This paper makes an important contribution to the study of decision-making under time pressure. The authors provide convincing evidence that decision boundaries can be highly nontrivial – even reaching infinity in realistic regimes. This paper will be of broad interest to both experimentalists and theorists working on decision-making under time pressure.

## Introduction

Even simple decisions can require us to adapt to a changing world. Should you go through the park or through town on your walk? The answer can depend on conditions that could be changing while you deliberate, such as an unexpected shower that would send you hurrying down the faster route (*Figure 1A*) or a predictable sunrise that would nudge you toward the route with better views. Despite the ubiquity of such dynamics in the real world, they are often neglected in models used to understand how the brain makes decisions. For example, many commonly used models assume that decision commitment occurs when the accumulated evidence for an option reaches a fixed, predefined value or threshold (*Wald, 1945*; *Ratcliff, 1978*; *Bogacz et al., 2006*; *Gold and*

**eLife digest** How do we make good choices? Should I have cake or yoghurt for breakfast? The strategies we use to make decisions are important not just for our daily lives, but also for learning more about how the brain works.

Decision-making strategies have two components: first, a deliberation period (when we gather information to determine which choice is 'best'); and second, a decision 'rule' (which tells us when to stop deliberating and commit to a choice). Although deliberation is relatively well-understood, less is known about the decision rules people use, or how those rules produce different outcomes.

Another issue is that even the simplest decisions must sometimes adapt to a changing world. For example, if it starts raining while you are deciding which route to walk into town, you would probably choose the driest route – even if it did not initially look the best. However, most studies of decision strategies have assumed that the decision-maker's environment does not change during the decision process.

In other words, we know much less about the decision rules used in real-life situations, where the environment changes. Barendregt et al. therefore wanted to extend the approaches previously used to study decisions in static environments, to determine which decision rules might be best suited to more realistic environments that change over time.

First, Barendregt et al. constructed a computer simulation of decision-making with environmental changes built in. These changes were either alterations in the quality of evidence for or against a particular choice, or the 'reward' from a choice, i.e., feedback on how good the decision was. They then used the computer simulation to model single decisions where these changes took place.

These virtual experiments showed that the best performance – for example, the most accurate decisions – resulted when the threshold for moving from deliberation (i.e., considering the evidence) to selecting an option could respond to, or even anticipate, the changing situations. Importantly, the simulations' results also predicted real-world choices made by human participants when given a decision-making task with similar variations in evidence and reward over time. In other words, the virtual decision-making rules could explain real behavior.

This study sheds new light on how we make decisions in a changing environment. In the future, Barendregt et al. hope that this will contribute to a broader understanding of decision-making and behavior in a wide range of contexts, from psychology to economics and even ecology.

---

*Shadlen, 2007*; *Kilpatrick et al., 2019*). The value of this threshold can account for inherent trade-offs between decision speed and accuracy found in many tasks: lower thresholds generate faster, but less accurate decisions, whereas higher thresholds generate slower, but more accurate decisions (*Gold and Shadlen, 2007*; *Chittka et al., 2009*; *Bogacz et al., 2010*). However, these classical models do not adequately describe decisions made in environments with potentially changing contexts (*Thura et al., 2014*; *Thura and Cisek, 2016*; *Palestro et al., 2018*; *Cisek et al., 2009*; *Drugowitsch et al., 2012*; *Thura et al., 2012*; *Tajima et al., 2019*; *Glickman et al., 2022*). Efforts to model decision-making thresholds under dynamic conditions have focused largely on heuristic strategies that aim to account for contexts that change between each decision. For instance, a common class of heuristic models is 'urgency-gating models' (UGMs). UGMs filter accumulated evidence through a low-pass filter and use thresholds that collapse monotonically over time (equivalent to dilating the belief in time) to explain decisions based on time-varying evidence quality (*Cisek et al., 2009*; *Carland et al., 2015*; *Evans et al., 2020*). Although collapsing decision thresholds are optimal in some cases, they do not always account for changes that occur during decision deliberation, and they are sometimes implemented ad-hoc without a proper derivation from first principles. Such derivations typically assume that individuals set decision thresholds to maximize trial-averaged reward rate (*Simen et al., 2009*; *Balci et al., 2011*; *Drugowitsch et al., 2012*; *Tajima et al., 2016*; *Malhotra et al., 2018*; *Boehm et al., 2020*), which can result in adaptive, time-varying thresholds similar to those assumed by heuristic UGMs. However, as in fixed-threshold models, these time-varying thresholds are typically defined before the evidence is accumulated, preceding the formative stages of the decision, and thus cannot account for environmental changes that may occur during deliberation.

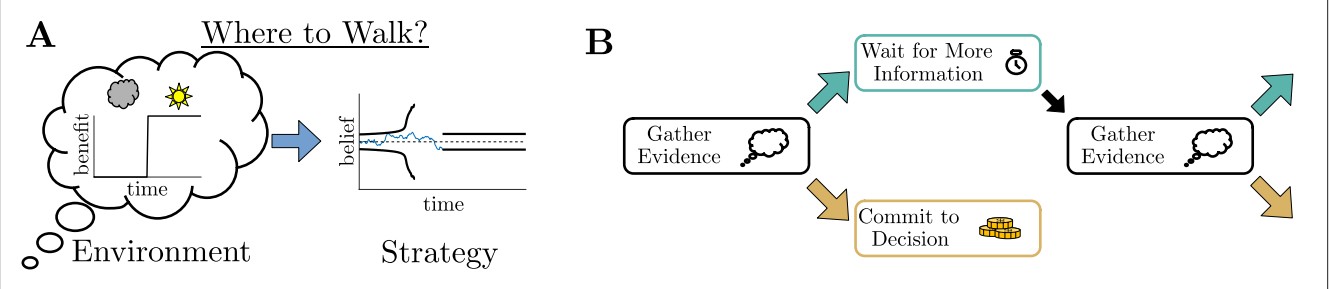

**Figure 1.** Simple decisions may require complex strategies. (**A**) When choosing where to walk, environmental fluctuations (e.g., weather changes) may necessitate changes in decision bounds (black line) adapted to changes in the conditions (cloudy to sunny). (**B**) Schematic of a dynamic programming. By assigning the best action to each moment in time, dynamic programming optimizes trial-averaged reward rate to produce the normative thresholds for a given decision.

To identify how environmental changes during the course of a single deliberative decision impact optimal decision rules, we developed normative models of decision-making that adapt to and anticipate two specific types of context changes: changes in reward expectation and changes in evidence quality. Specifically, we used Bellman's equation (***Bellman, 1957***; ***Mahadevan, 1996***; ***Sutton and Barto, 1998***; ***Bertsekas, 2012***; ***Drugowitsch, 2015***) to identify decision strategies that maximize trial-averaged reward rate when conditions can change during decision deliberation. We show that for simple tasks that include sudden, expected within-trial changes in the reward or the quality of observed evidence, these normative decision strategies involve non-trivial, time-dependent changes in decision thresholds. These rules take several different forms that outperform their heuristic counterparts, are identifiable from behavior, and have performance that is robust to noisy implementations. We also show that, compared to fixed-threshold models or UGMs, these normative, adaptive threshold models provide a better account of human behavior on a 'tokens task', in which the value of commitment changes gradually at predictable times and the quality of evidence changes unpredictably within each trial (***Cisek et al., 2009***; ***Thura et al., 2014***). These results provide new insights into the behavioral relevance of a diverse set of adaptive decision thresholds in dynamic environments and tightly link the details of such environmental changes to threshold adaptations.

## Results

### Normative theory for dynamic context 2AFC tasks

To determine potential deliberation and commitment strategies used by human subjects, we begin by identifying normative decision rules for two-alternative forced choice (2AFC) tasks with dynamic contexts. Normative decision rules that maximize trial-averaged reward rate can be obtained by solving an optimization problem using dynamic programming (***Bellman, 1957***; ***Sutton and Barto, 1998***; ***Drugowitsch et al., 2012***; ***Tajima et al., 2016***). We define this trial-averaged reward rate, $\rho$, as (***Gold and Shadlen, 2002***; ***Drugowitsch et al., 2012***)

$$\rho = \frac{\langle R \rangle - \langle C(T_d) \rangle}{\langle T_t \rangle + \langle t_i \rangle}, \tag{1}$$

where $\langle R \rangle$ is the average reward for a decision, $T_d$ is the decision time, $\langle C(T_d) \rangle = \left\langle \int_0^{T_d} c(t)\, dt \right\rangle$ is the average total accumulated cost given an incremental cost function $c(t)$, $\langle T_t \rangle$ is the average trial length, and $\langle t_i \rangle$ is the average inter-trial interval (***Drugowitsch, 2015***). Note that all averages in *Equation 1* are taken over trials. To find the normative decision thresholds that maximize $\rho$, we assign specific values (i.e., economic utilities) to correct and incorrect choices (reward and/or punishment) and the time required to arrive at each choice (i.e., evidence cost). The incremental evidence function $c(t)$ represents both explicit time costs, such as a price for gathering evidence, and implicit costs, such as opportunity cost. While there are many forms of this cost function, we make the simplifying assumption that it is constant, $c(t) = c$. Because more complex cost functions can influence decision threshold dynamics (***Drugowitsch et al., 2012***), restricting the cost function to a constant ensures that the threshold dynamics we identify are governed purely by changes in the (external) task conditions and

not the (internal) cost function. To represent the structure of a 2AFC tasks, we assume a decision environment for an observer with an initially unknown environmental state, $s \in \{s_+, s_-\}$, that uniquely determines which of two alternatives is correct. To infer the environmental state, this observer makes measurements, $\xi$, that follow a distribution $f_\pm(\xi) = f(\xi|s_\pm)$ that depends on the state. Determining the correct choice is thus equivalent to determining the generating distribution, $f_\pm$. An ideal Bayesian observer uses the log-likelihood ratio (LLR), $y$, to track their 'belief' over the correct choice (**Wald, 1945**; **Bogacz et al., 2006**; **Veliz-Cuba et al., 2016**). After $n$ discrete observations $\xi_{1:n}$ that are independent across time, the discrete-time LLR belief $y_n$ is given by:

$$y_n = \ln \frac{\Pr(s_+|\xi_{1:n})}{\Pr(s_-|\xi_{1:n})} = \ln \frac{f_+(\xi_n)}{f_-(\xi_n)} + y_{n-1}. \tag{2}$$

Given this defined task structure, we discretize the time during which the decision is formed and define the observer's actions during each timestep. The observer gathers evidence (measurements) during each timestep prior to a decision and uses each increment of evidence to update their belief about the correct choice. Then, the observer has the option to either commit to a choice or make another measurement at the next timestep. By assigning a utility to each of these actions (i.e., a value $V_+$ for choosing $s_+$, a value $V_-$ for choosing $s_-$, and a value $V_w$ for sampling again), we can construct the value function for the observer given their current belief:

$$
\begin{aligned}
V(p_n; \rho) &= \max\{V_+(p_n; \rho), V_-(p_n; \rho), V_w(p_n; \rho)\} \\
&= \max \begin{cases} R_c p_n + R_i (1 - p_n) - \langle t_i \rangle \rho, & \text{choose } s_+ \\ R_c (1 - p_n) + R_i p_n - \langle t_i \rangle \rho, & \text{choose } s_- \\ \langle V(p_{n+1}; \rho)|p_n \rangle_{p_{n+1}} - c(t)\delta t - \rho \delta t, & \text{sample again} \end{cases}
\end{aligned}
\tag{3}
$$

For a full derivation of this equation, see Materials and methods. In **Equation 3**, $p_n = \Pr(s_+|\xi_{1:n}) = \frac{1}{1+e^{-y_n}}$ is the state likelihood at time $t_n$, $R_c$ is the reward for a correct choice, $R_i$ is the reward for an incorrect choice, and $\delta t$ is the timestep between observations. We choose generating distributions to be symmetric Gaussian distributions $f_\pm(\xi) \sim \mathcal{N}\left(\pm\mu, \sigma^2\right)$ to allow us to compute the conditional distribution function $f_p(p_{n+1}|p_n)$ needed for the average future value explicitly:

$$\langle V(p_{n+1}; \rho) \,|\, p_n \rangle_{p_{n+1}} = \int_0^1 V(p_{n+1}; \rho) f_p(p_{n+1} | p_n) \, d p_{n+1}. \tag{4}$$

In **Equation 4**, $f_p(p_{n+1}|p_n)$ is the conditional probability of the future state likelihood $p_{n+1}$ given the current state likelihood $p_n$. For the case of Gaussian-distributed evidence, this conditional probability is given by **Equation 16** in Materials and methods. Using **Equation 3**, we find the specific belief values where the optimal action changes from gathering evidence to commitment, defining thresholds on the ideal observer's belief that trigger decisions. **Figure 1B** shows a schematic of this process.

To understand how normative decision thresholds adapt to changing conditions, we derived them for several different forms of two-alternative forced-choice (2AFC) tasks in which we controlled changes in evidence or reward. Even for such simple tasks, there is a broad set of possible changing contexts. In the next section, we analyze a task in which context changes gradually (the tokens task). Here, we focus on tasks in which the context changes abruptly. For each task, an ideal observer was shown evidence generated from a Gaussian distribution $f_\pm(\xi) = \mathcal{N}\left(\pm\mu, \sigma^2\right)$ with signal-to-noise ratio (SNR) $m = \frac{2\mu^2}{\sigma^2}$ (**Figure 2—figure supplement 1**). The SNR measures evidence quality: a smaller (larger) $m$ implies that evidence is of lower (higher) quality, resulting in harder (easier) decisions. The observer's goal was to determine which of the two means (i.e., which distribution, $f_+$ or $f_-$) were used to generate the observations. We introduced changes in the reward for a correct decision ('reward-change task') or the SNR ('SNR-change task') within a single decision, where the time and magnitude of the changes are known in advance to the observer (**Figure 1A**, **Figure 2—figure supplement 2**). For example, changes in SNR arise naturally throughout a day as animals choose when to forage and hunt given variations in light levels and therefore target-acquisition difficulty (**Combes et al., 2012**; **Einfalt et al., 2012**).

Under these dynamic conditions, dynamic programming produces normative thresholds with rich non-monotonic dynamics (**Figure 2A and B**, **Figure 2—figure supplement 2**). Environments with

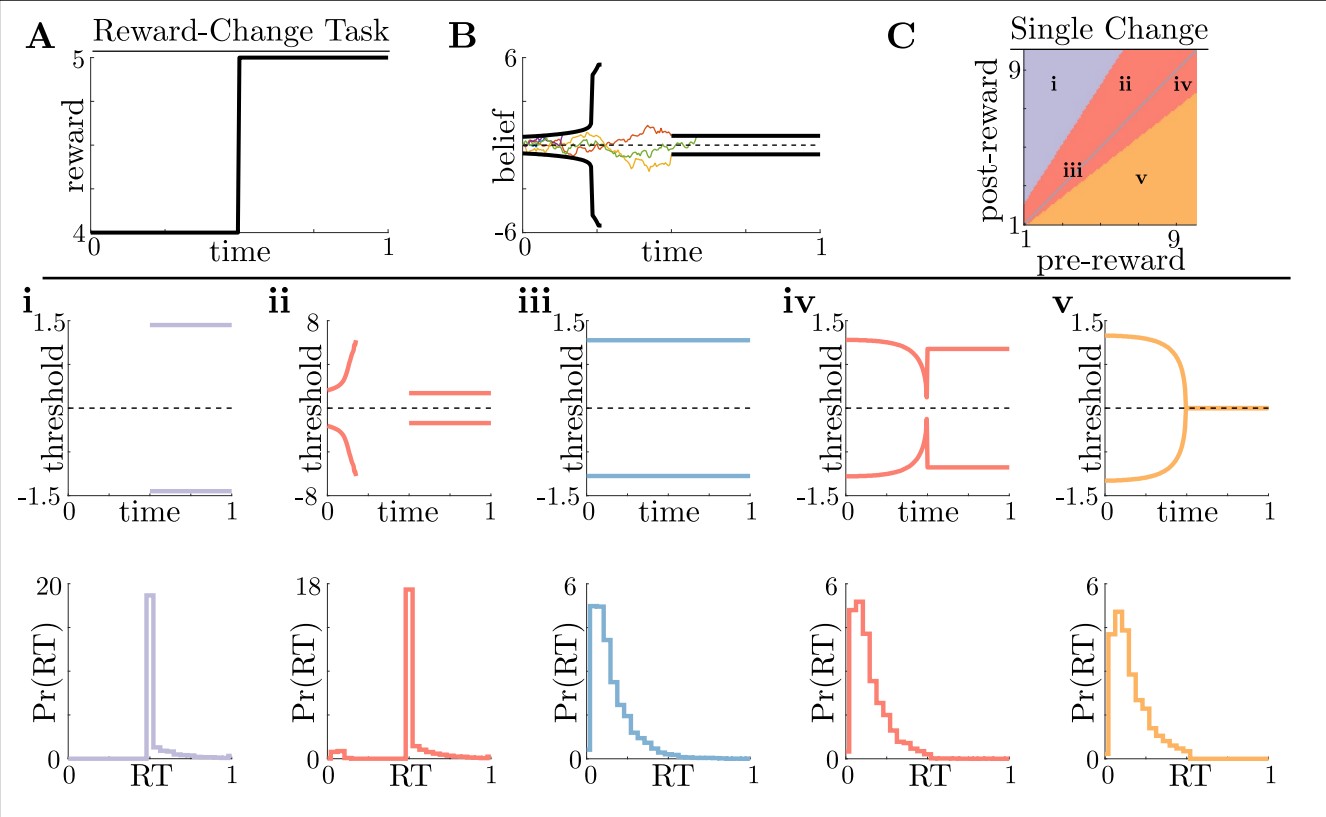

**Figure 2.** Normative decision rules are characterized by non-monotonic task-dependent motifs. (**A,B**) Example reward time series for a reward-change task (black lines in A), with corresponding thresholds found by dynamic programming (black lines in B). The colored lines in B show sample realizations of the observer's belief. (**C**) To understand the diversity of threshold dynamics, we consider the simple case of a single change in the reward schedule. The panel shows a colormap of normative threshold dynamics for these conditions. Distinct threshold motifs are color-coded, corresponding to examples shown in panels i-v. (**i-v**): Representative thresholds (top) and empirical response distributions (bottom) from each region in C. During times at which thresholds in the upper panels are not shown (e.g., $t \in [0, 0.5]$ in i), the thresholds are infinite and the observer will never respond. For all simulations, we take the incremental cost function $c(t) = 1$, punishment $R_i = 0$, evidence quality $m = 5$, and inter-trial interval $t_i = 1$.

The online version of this article includes the following figure supplement(s) for figure 2:

**Figure supplement 1.** Impact of evidence quality on belief and difficulty.

**Figure supplement 2.** Normative thresholds for reward-change task with multiple changes.

**Figure supplement 3.** Threshold dynamics in the inferred reward-change task.

multiple reward changes during a single decision lead to complex threshold dynamics that we summarize in terms of several threshold change "motifs." These motifs occur on shorter intervals and tend to emerge from simple monotonic changes in context parameters (**Figure 2—figure supplement 2**). To better understand the range of possible threshold motifs, we focused on environments with single changes in task parameters. For the reward-change task, we set punishment $R_i = 0$ and assumed reward $R_c$ changes abruptly, so that its dynamics are described by a Heaviside function:

$$R_c(t) = (R_2 - R_1)H_\theta(t - 0.5) + R_1. \tag{5}$$

Thus, the reward switches from the pre-change reward $R_1$ to the post-change reward $R_2$ at $t = 0.5$.

For this single-change task, normative threshold dynamics exhibited several motifs that in some cases resembled fixed or collapsing thresholds characteristic of previous decision models but in other cases exhibited novel dynamics. Specifically, we characterized five different dynamic motifs in response to single, expected changes in reward contingencies for different combinations of pre- and post-change reward values (**Figure 2C and i–v**). For tasks in which reward is initially very low, thresholds are infinite until the reward increases, ensuring that the observer waits for the larger payout regardless of how strong their belief is (**Figure 2i**). The region where thresholds are infinite corresponds to when

$V_w(p_n; \rho)$ in *Equation 3*, which is the value associated with waiting to gather more information, is maximal for all values of $p_n$. In contrast, when reward is initially very high, thresholds collapse to zero just before the reward decreases, ensuring that all responses occur while payout is high (*Figure 2v*). Between these two extremes, optimal thresholds exhibit rich, non-monotonic dynamics (*Figure 2ii,iv*), promoting early decisions in the high-reward regime, or preventing early, inaccurate decisions in the low-reward regime. *Figure 2C* shows the regions in pre- and post-change reward space where each motif is optimal, including broad regions with non-monotonic thresholds. Thus, even simple context dynamics can evoke complex decision strategies in ideal observers that differ from those predicted by constant decision-thresholds and heuristic UGMs.

We also formulated an 'inferred reward-change task', in which reward fluctuates between a high value $R_H$ and low value $R_L$ governed by a two-state Markov process with known transition rate $h$ and state $R(t) \in \{R_H, R_L\}$ that the observer must infer on-line. For this task, the observer receives two independent sets of evidence: the evidence of the state $\xi|s_\pm \sim \mathcal{N}\left(\pm\mu, \sigma^2\right)$ and the evidence of the current reward $\eta|R_{H/L} \sim \mathcal{N}\left(\pm\mu_R, \sigma_R^2\right)$. The observer must then track their beliefs about both the state and the current reward and take both sources of information into account when determining the optimal decision thresholds. We found that these thresholds always changed monotonically with monotonic shifts in expected reward (see *Figure 2—figure supplement 3*). These results contrast with our findings from the reward-change task in which changes can be anticipated and monotonic changes in reward can produce non-monotonic changes in decision thresholds.

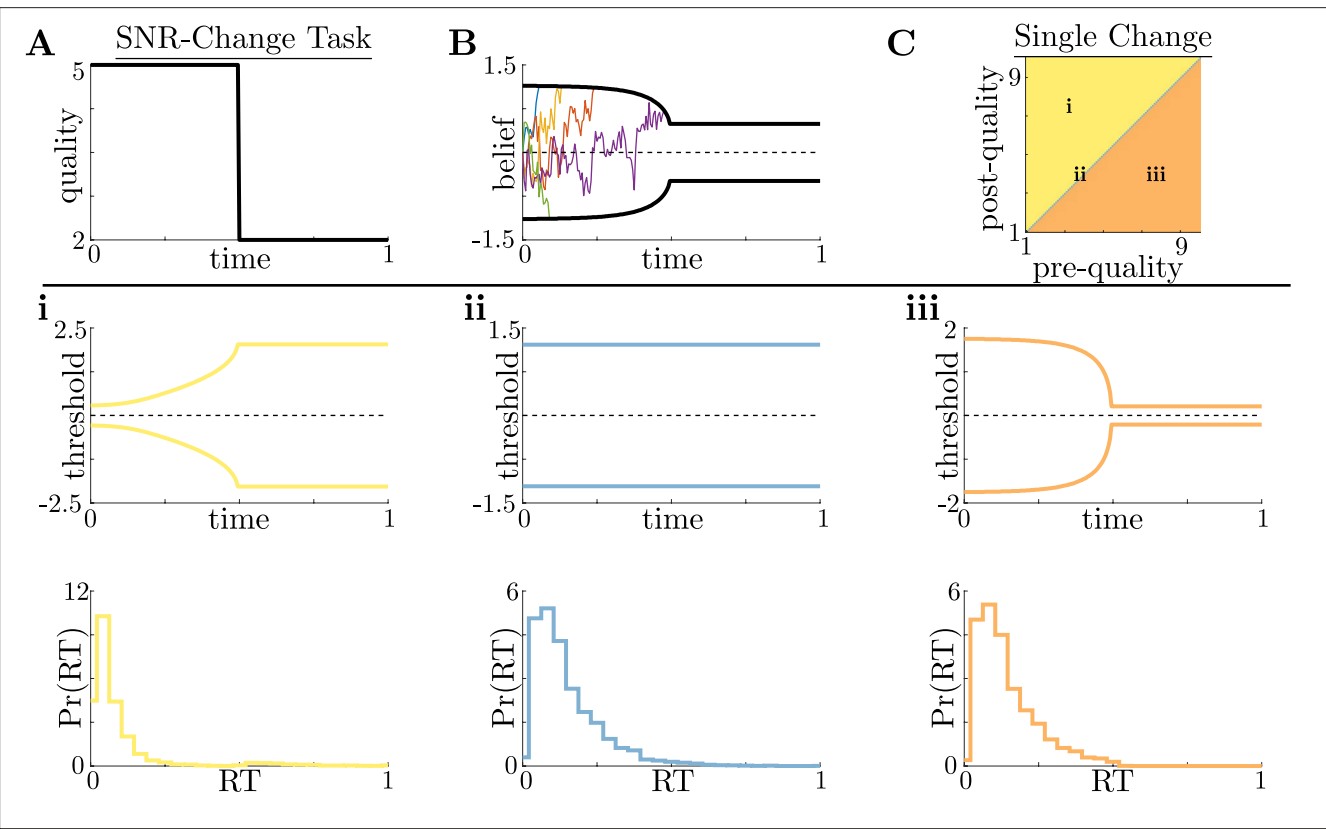

**Figure 3.** Dynamic-quality task does not exhibit non-monotonic motifs. (**A,B**) Example quality time series for the SNR-change task (**A**), with corresponding thresholds found by dynamic programming (**B**). Colored lines in B show sample realizations of the observer's belief. As in *Figure 2*, we characterize motifs in the threshold dynamics and response distributions based on single changes in SNR. (**C**) Colormap of normative threshold dynamics for a known reward schedule task with a single quality change. Distinct dynamics are color-coded, corresponding to examples shown in panels i-iii. (**i-iii**): Representative thresholds (top) and empirical response distributions (bottom) from each region in C. For all simulations, we take the incremental cost function $c(t) = 1$, reward $R_c = 5$, punishment $R_i = 0$, and inter-trial interval $t_i = 1$.

The online version of this article includes the following figure supplement(s) for figure 3:

**Figure supplement 1.** Normative thresholds for SNR-change task with multiple changes.

For the SNR-change task, optimal strategies for environments with multiple changes in evidence quality are characterized by threshold dynamics that adapt to these changes in a way similar to how they adapt to changes in reward (*Figure 3—figure supplement 1*). To study the range of possible threshold motifs, we again considered environments with single changes in the evidence quality $m = \frac{2\mu^2}{\sigma^2}$ by taking μ to be a Heaviside function:

$$\mu(t) = (\mu_2 - \mu_1)H_\theta(t - 0.5) + \mu_1. \tag{6}$$

For this single-change task, we again found similar threshold motifs to those in the reward-change task (*Figure 3A and B*). However, in this case monotonic changes in evidence quality always produce monotonic changes in response behavior. This observation holds across all of parameter space for evidence-quality schedules with single change points (*Figure 3C*), with only three optimal behavioral motifs (*Figure 3i–iii*). This contrasts with our findings in the reward-change task, where monotonic changes in reward can produce non-monotonic changes in decision thresholds. Strategies arising from known dynamical changes in context tend to produce sharper response distributions around reward changes than around quality changes, which may be measurable in psychophysical studies. These findings suggest that changes in reward can have a larger impact on the normative strategy thresholds than changes in evidence quality.

## Performance and robustness of non-monotonic normative thresholds

The normative solutions that we derived for dynamic-context tasks by definition maximize reward rate. This maximization assumes that the normative solutions are implemented perfectly. However, a perfect implementation may not be possible, given the complexity of the underlying computations, biological constraints on computation time and energy (*Louie et al., 2015*), and the synaptic and neural variability of cortical circuits (*Ma and Jazayeri, 2014*; *Faisal et al., 2008*). Given these constraints, subjects may employ heuristic strategies like the UGM over the normative model if noisy or mistuned versions of both models result in similar reward rates. We used synthetic data to better understand the relative benefits of different imperfectly implemented strategies. Specifically, we corrupted the internal belief state and simulated response times with additive Gaussian noise with zero mean and variance $\sigma_{mn}^2$ (See *Figure 4—figure supplement 1C*) for three models:

1. The continuous-time normative model with time-varying thresholds $\pm\theta(t)$ from *Equation 3* and belief that evolves according to the stochastic differential equation

$$d\tilde{y} = \underbrace{\pm m\,dt}_{\text{drift}} + \underbrace{\sqrt{2m}\,dW_t}_{\text{sample noise}} + \underbrace{\sigma_y\,dW'_t}_{\text{sensory noise}},$$

   where $dW_t$ is a standard increment of a Wiener process, the sign of the drift $\pm m\,dt$ is given by the correct choice $s_\pm$, and $dW'_t$ is an independent Wiener process with strength $\sigma_y$. The addition of the additional noise process $dW'_t$ makes this a noisy Bayesian (NB) model.
2. A constant-threshold (Const) model, which uses the same belief $\tilde{y}$ as the normative model but a constant, non-adaptive decision threshold $\pm\theta(t) = \pm\theta_0$ (*Figure 4—figure supplement 1A*).
3. The UGM, which uses the output of a low-pass filter as the belief,

$$\tau\,dE = \underbrace{\left(-E + \frac{1}{1 + e^{-y}} - \frac{1}{2}\right)dt}_{\text{drift \& sample noise}} + \underbrace{\sigma_y\,dW_t}_{\text{sensory noise}}, \tag{7}$$

   and commits to a decision when this output crosses a hyperbolically collapsing threshold $\pm\theta(t) = \pm\frac{\theta_0}{at}$ (*Figure 4—figure supplement 1B*). In *Equation 7*, $E$ is the filter's output that serves as the UGM's belief, $\tau$ is a relaxation time constant, and the optimal observer's belief $y$ is the filter's input. Note that the filter's input can also be written in terms of the state likelihood $p$,

$$\tau\,dE = \left(-E + p - \frac{1}{2}\right)dt + \sigma_y\,dW_t,$$

   which is the form first proposed by *Cisek et al., 2009*.

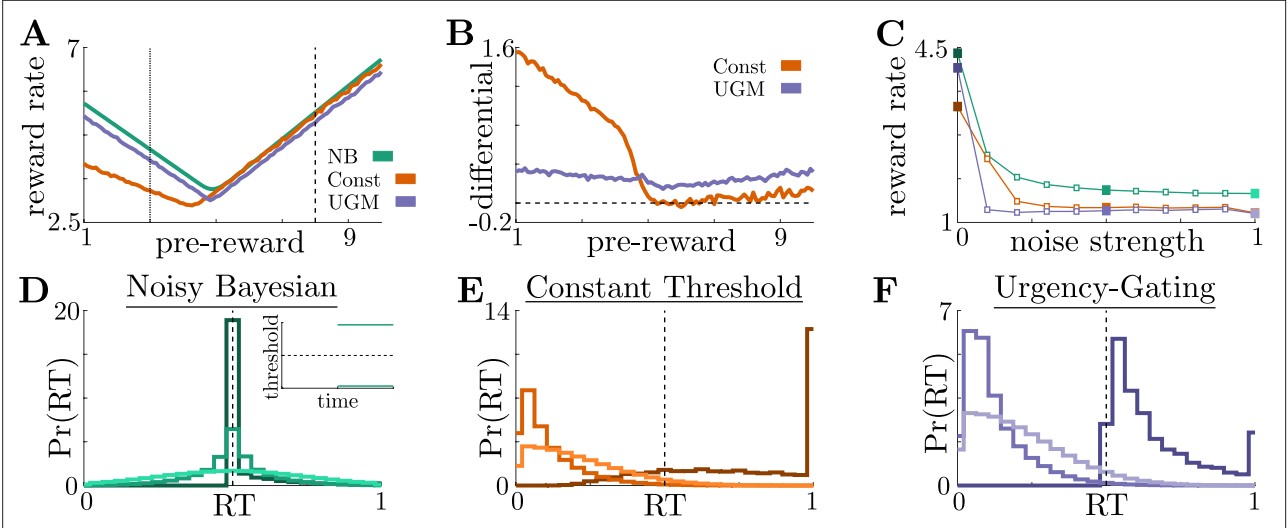

**Figure 4.** Benefits of adaptive normative thresholds compared to heuristics. (**A**) Reward rate $\rho$ for the noisy Bayesian (NB) model, constant-threshold (Const) model, and UGM for the reward-change task, where all models are tuned to maximize performance with zero sensory noise ($\sigma_y = 0$) and zero motor noise ($\sigma_{mn} = 0$); in this case, the NB model is equivalent to the optimal normative model. Model reward rates are shown for different pre-change rewards $R_1$, with post-change reward $R_2$ set so that $R_1 + R_2 = 11$ to keep the average total reward fixed (see Materials and methods for details). Low-to-high reward changes (dotted line) produce larger performance differentials than high-to-low changes (dashed line). (**B**) Absolute reward rate differential between NB and alternative models, given by $\rho(\text{NB}) - \rho(\text{alt})$ for different pre-change rewards. Legend shows which alternate model was used to produce each curve. (**C**) Reward rates of all models for reward-change task with $(R_1, R_2) = (3, 8)$ as both observation and response-time noise is increased. Noise strength for each model is given by $\frac{\sigma_y + \sigma_{mn}}{\overline{\sigma}_y + \overline{\sigma}_{mn}}$, are $\overline{\sigma}_y = 5$ and $\overline{\sigma}_{mn} = 0.25$ were the maximum strengths of $\sigma_y$ and $\sigma_{mn}$ we considered (See *Figure 4—figure supplement 1C* for reference). Filled markers correspond to no noise, moderate noise, and high noise strengths. D,E,F: Response distributions for (**D**) NB; (**E**) Const; and (**F**) UGM models in a low-to-high reward environment with $(R_1, R_2) = (3, 8)$. In each panel, results derived for several noise strengths, corresponding with filled markers in C, are superimposed, with lighter distributions denoting higher noise. Inset in D shows normative thresholds obtained from dynamic programming. Dashed line shows time of reward increase. For all simulations, we take the incremental cost function $c(t) = 1$, punishment $R_i = 0$, and evidence quality $m = 5$.

The online version of this article includes the following figure supplement(s) for figure 4:

**Figure supplement 1.** Heuristic model and noise schematics.

**Figure supplement 2.** Model performance for high-to-low reward switch.

**Figure supplement 3.** Model performance for decomposed noise strengths.

For more details about these three models, see Materials and methods. We compared their performance in terms of reward rate achieved on the same set of reward-change tasks shown in *Figure 2*. To ensure the average total reward in each trial was the same, we restricted the pre-change reward $R_1$ and post-change reward $R_2$ so that $R_1 + R_2 = 11$.

When all three models were implemented without additional noise, the relative benefits of the normative model depended on the exact task condition. The performance differential between models was highest when reward changed from low to high values (*Figure 4A*, dotted line; *Figure 4*). Under these conditions, normative thresholds are initially infinite and become finite after the reward increases, ensuring that most responses occur immediately once the high reward becomes available (*Figure 4D*). In contrast, response times generated by the constant-threshold and UGM models tend to not follow this pattern. For the constant-threshold model, many responses occur early, when the reward is low (*Figure 4E*). For the UGM, a substantial fraction of responses are late, leading to higher time costs however, it is possible to tune the UGM's thresholds rate of collapse to prevent any early responses while the reward is low (*Figure 4F*). In contrast, when the reward changes from high to low values, all models exhibit similar response distributions and reward rates (*Figure 4A*, dashed line; *Figure 4—figure supplement 2*). This result is not surprising, given that the constant-threshold model produces early peaks in the reaction time distribution, and the UGM was designed to mimic collapsing bounds that hasten decisions in response to imminent decreases in reward (*Cisek et al., 2009*). We therefore focused on the robustness of each strategy when corrupted by noise and responding to

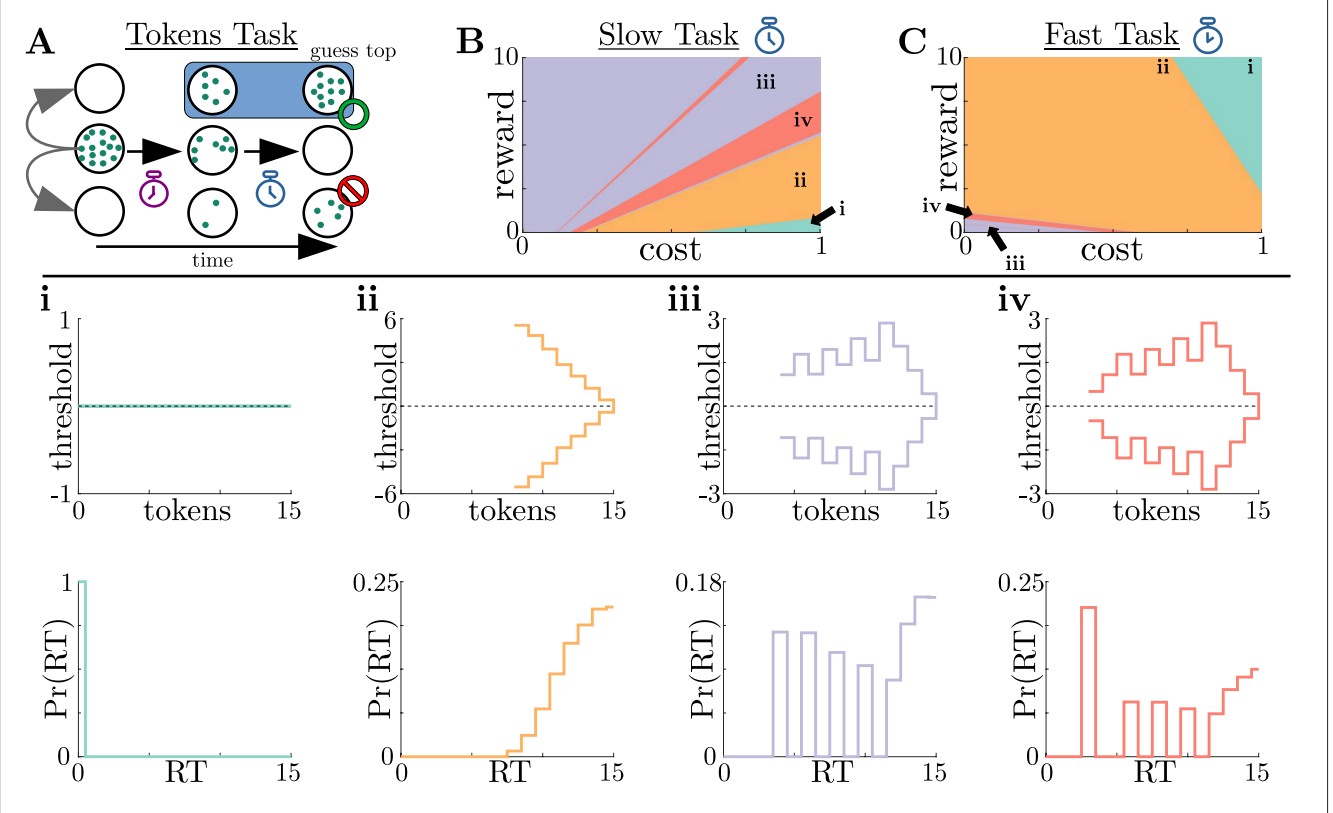

**Figure 5.** Normative strategies for the tokens task exhibit various distinct decision threshold motifs with sharp, non-monotonic changes. (**A**) Schematic of the tokens task. The subject must predict which target (top or bottom) will have the most tokens once all tokens have left the center target (see text for details). (**B**) Colormap of normative threshold dynamics for the 'slow' version of the tokens task in reward-evidence cost parameter space (i.e., as a function of $R_c$ and $c(t) = c$ from *Equation 3*, with punishment $R_i$ set to –1). Distinct dynamics are color-coded, with different motifs shown in i-iv. (**C**) Same as B, but for the 'fast' version of the tokens task. (**i-iv**): Representative thresholds (top) and empirical response distributions (bottom) from each region in (**B,C**). Thresholds are plotted in the LLR-belief space $y_n = \ln \frac{p_n}{1-p_n}$, where $p_n$ is the state likelihood given by *Equation 8*. Note that we distinguish iii and iv by the presence of either one (iii) or multiple (iv) consecutive threshold increases. In regions where thresholds are not displayed (e.g., $N_t \in \{0, \dots, 7\}$ in ii), the thresholds are infinite.

The online version of this article includes the following figure supplement(s) for figure 5:

**Figure supplement 1.** Tokens task thresholds in token lead space.

low-to-high reward switches – the regime differentiating strategy performance in ways that could be identified in subject behavior.

Adding noise to the internal belief state (which tends to trigger earlier responses) and simulated response distributions (which tends to smooth out the distributions) without re-tuning the models to account for the additional noise does not alter the advantage of the normative model: across a range of added noise strengths, which we define as $\frac{\sigma_y + \sigma_{mn}}{\overline{\sigma}_y + \overline{\sigma}_{mn}}$, where $\overline{\sigma}_y$ and $\overline{\sigma}_{mn}$ are the maximum possible strengths of sensory and motor noise, respectively, the normative model outperforms the other two when encountering low-to-high reward switches (*Figure 4C*). This robustness arises because, prior to the reward change, the normative model uses infinite decision thresholds that prevent early noise-triggered responses when reward is low (*Figure 4D*). In contrast, the heuristic models have finite collapsing or constant thresholds and thus produce more suboptimal early responses as belief noise is increased (*Figure 4E and F*). Thus, adaptive decision strategies can result in considerably higher reward rates than heuristic alternatives even when implemented imperfectly, suggesting subjects may be motivated to learn such strategies.

## Adaptive normative strategies in the tokens task

To determine the relevance of the normative model to human decision-making, we analyzed previously collected data from a 'tokens task' (*Cisek et al., 2009*). For this task, human subjects were

shown 15 tokens inside a center target flanked by two empty targets (see *Figure 5A* for a schematic). Every 200ms, a token moved from the center target to one of the neighboring targets with equal probability. Subjects were tasked with predicting which flanking target would contain more tokens by the time all 15 moved from the center. Subjects could respond at any time before all 15 tokens had moved. Once the subject made the prediction, the remaining tokens would finish their movements to indicate the correct alternative. Given this task structure, one can show using a combinatorial argument (*Cisek et al., 2009*) that the state likelihood function $p_n = \Pr(\text{top} | \xi_{1:n})$, the probability the top target will hold more tokens at the end of the trial, is given by:

$$p_n = p(\text{top} | U_n, L_n, C_n) = \frac{C_n!}{2^{C_n}} \sum_{k=0}^{\min\{C_n, 7-L_n\}} \frac{1}{k!(C_n - k)!}, \tag{8}$$

where $U_n$, $L_n$, and $C_n$ are the number of tokens in the upper, lower, and center targets after token movement $n$, respectively. The token movements are Markovian because each token has an equal chance of moving to the upper/lower target. However, the probability that a target will contain more tokens at the end of the trial is history dependent, and the evolution of these probabilities is thus non-Markovian. As such, the quality of evidence possible from each token draw changes dynamically and gradually. In addition, the task included two different post-decision token movement speeds, 'slow' and 'fast': once the subject committed to a choice, the tokens finished out their animation, moving either once every 170ms (slow task) or once every 20ms (fast task). This post-decision movement acceleration changed the value associated with commitment by making the average inter-trial interval ($\langle t_i \rangle$ in *Equation 1*) decrease over time. Because of this modulation, we can interpret the tokens task as a multi-change reward task, where commitment value is controlled through $\langle t_i \rangle$ rather than through reward $R_c$. Our dynamic-programming framework for generating adaptive decision rules can handle the gradual changes in task context emerging in the tokens task. Given that costs and rewards can be subjective, we quantified how normative decision thresholds change with different combinations of rewards $R_c$ and costs $c(t) = c$ for fixed punishment $R_i = -1$, for both the slow (*Figure 5B*) and fast (*Figure 5C*) versions of the task.

We identified four distinct motifs of normative decision threshold dynamics for the tokens task (*Figure 5i-iv*). Some combinations of rewards and costs produced collapsing thresholds (*Figure 5ii*) similar to the UGM developed by *Cisek et al., 2009* for this task. In contrast, large regions of task parameter space produced rich non-monotonic threshold dynamics (*Figure 5iii,iv*) that differed from any found in the UGM. In particular, as in the case of reward-change tasks, normative thresholds were often infinite for the first several token movements, preventing early and weakly informed responses. These motifs are similar to those produced by low-to-high reward switches in the reward-change task, but here resulting from the low relative cost of early observations. These non-monotonic dynamics also appear if we measure belief in terms of the difference in tokens between the top and bottom target, which we call 'token lead space' (see *Figure 5—figure supplement 1*).

## Adaptive normative strategies best fit subject response data

To determine the relevance of these adaptive decision strategies to human behavior, we fit discrete-time versions of the noisy Bayesian (four free parameters), constant-threshold (three free parameters), and urgency-gating (five free parameters) models to response-time data from the tokens task collected by *Cisek et al., 2009*; see Table 1 in Materials and methods for a table of parameters for each model. All models included belief and motor noise, as in our analysis of the dynamic-context tasks (*Figure 4—figure supplement 1C*). The normative model tended to fit the data better than the heuristic models (see *Figure 6—figure supplement 1*), based on three primary analyses. First, both corrected AIC (AICc), which accounts for goodness-of-fit and model degrees-of-freedom, and average root-mean-squared error (RMSE) between the predicted and actual trial-by-trial response times, favored the noisy Bayesian model for most subjects for both the slow (*Figure 6A*) and fast (*Figure 6D*) versions of the task. Second, when considering only the best-fitting model for each subject and task condition, the noisy Bayesian model tended to better predict subject's response times (*Figure 6B and E*). Third, most subjects whose data were best described by the noisy Bayesian model had best-fit parameters that corresponded to non-monotonic decision thresholds, which cannot be produced by either of the other two models (*Figure 6C and F*). This result also shows that, assuming subjects used a normative

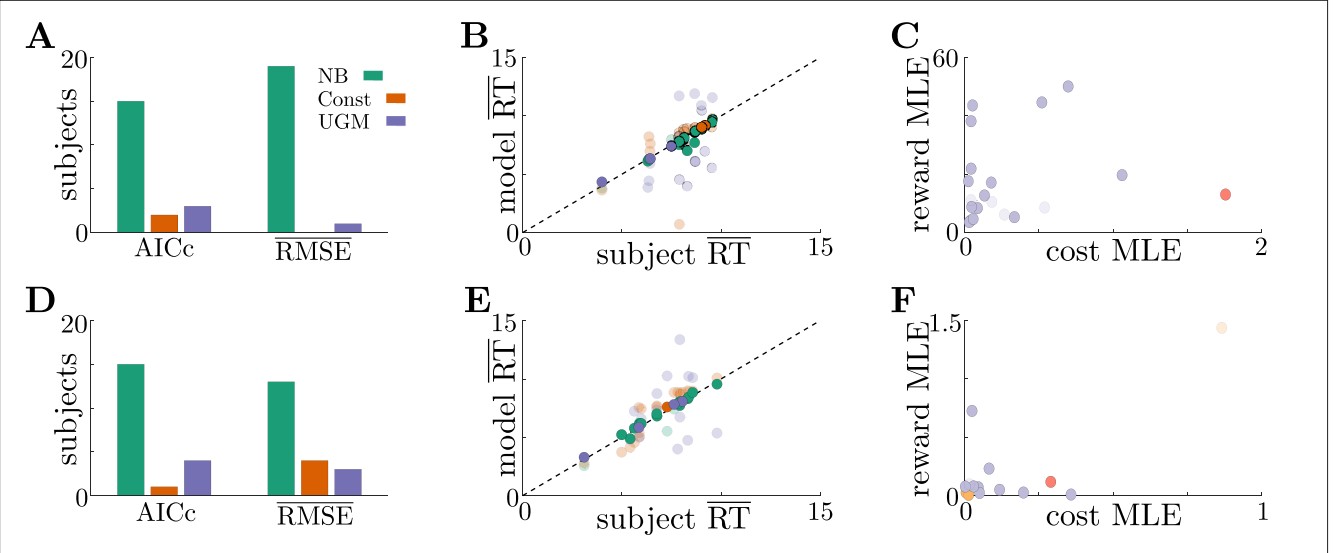

**Figure 6.** Adaptive normative strategies provide the best fit to subject behavior in the tokens task. (**A**) Number of subjects from the slow version of the tokens task whose reponses were best described by each model (legend) identified using corrected AIC (left) and average trial-by-trial RMSE (right). (**B**) Comparison of mean RT from subject data in the slow version of the tokens task ($x$-axis) to mean RT of each fit model ($y$-axis) at maximum-likelihood parameters. Each symbol is color-coded to agree with its associated model. Darker symbols correspond to the model that best describes the responses of a subject selected using corrected AIC. The NB model had the lowest variance in the difference between predicted and measured mean RT (NB var: 0.13, Const var: 3.11, UGM var: 5.39). (**C**) Scatter plot of maximum-likelihood parameters $R_c$ and $c(t) = c$ for the noisy Bayesian model for each subject in the slow version of the task. Each symbol is color-coded to match the threshold dynamics heatmap from *Figure 5B*. Darker symbols correspond to subjects whose responses were best described by the noisy Bayesian model using corrected AIC. (**D-F**) Same as A-C, but for the fast version of the tokens task. The NB model had the lowest variance in the difference between predicted and measured mean RT in this version of the task (NB var: 0.22, Const var: 0.82, UGM var: 5.32).

The online version of this article includes the following figure supplement(s) for figure 6:

**Figure supplement 1.** Summary of model fits.

**Figure supplement 2.** Fitted posterior for each model parameter for NB model.

model, they used distinct model parameters, and thus different strategies, for both the fast and slow task conditions. This finding is clearer when looking at the posterior parameter distribution for each subject and model parameter (see *Figure 6—figure supplement 1* for an example). We speculate that the higher estimated value of reward in the slow task may arise due to subjects valuing frequent rewards more favorably. Together, our results strongly suggest that these human subjects tended to use an adaptive, normative strategy instead of the kinds of heuristic strategies often used to model response data from dynamic context tasks.

## Discussion

The goal of this study was to build on previous work showing that in dynamic environments, the most effective decision processes do not necessarily use relatively simple, pre-defined computations as in many decision models (*Bogacz et al., 2006*; *Cisek et al., 2009*; *Drugowitsch et al., 2012*), but instead adapt to learned or predicted features of the environmental dynamics (*Drugowitsch et al., 2014a*). Specifically, we used new 'dynamic context' task structures to demonstrate that normative decision commitment rules (i.e., decision thresholds, or bounds, in 'accumulate-to-bound' models) adapt to reward and evidence-quality switches in complex, but predictable, ways. Comparing the performance of these normative decision strategies to the performance of classic heuristic models, we found that the advantage of normative models is maintained when computations are noisy. We extended these modeling results to include the 'tokens task', in which evidence quality changes in a way that depends on stimulus history and the utility of commitment increases over time. We found that the normative decision thresholds for the tokens task are also non-monotonic and robust to noise. By reanalyzing human subject data from this task, we found most subjects' response times were best-explained by a

noisy normative model with non-monotonic decision thresholds. Taken collectively, these results show that ideal observers and human subjects use adaptive and robust normative decision strategies in relatively simple decision environments.

Our results can aid experimentalists investigating the nuances of complex decision-making in several ways. First, we demonstrated that normative behavior varies substantially across task parameters for relatively simple tasks. For example, the reward-change task structure produces five distinct behavioral motifs, such as waiting until reward increases (*Figure 2i*) and responding before reward decreases unless the accumulated evidence is ambiguous (*Figure 2iv*). Using these kinds of modeling results to inform experimental design can help us understand the possible behaviors to expect in subject data. Second, extending our work and considering the sensitivity of performance to both model choice and task parameters (*Barendregt et al., 2019*; *Radillo et al., 2019*) will help to identify regions of task parameter space where models are most identifiable from observables like response time and choice. Third, and more generally, our work provides evidence that for tasks with gradual changes in evidence quality and reward, human behavior is more consistent with normative principles than with previously proposed heuristic models. However, more work is needed to determine if and how people follow normative principles for other dynamic-context tasks, such as those involving abrupt changes in evidence or reward contingencies, by using normative theory to determine which subject strategies are plausible, the nature of tasks needed to identify them, and the relationship between task dynamics and decision rules.

Model-driven experimental design can aid in identification of adaptive decision rules in practice. People commonly encounter unpredictable (e.g. an abrupt thunderstorm) and predictable (e.g. sunset) context changes when making decisions. Natural extensions of common perceptual decision tasks (e.g. random-dot motion discrimination [*Gold and Shadlen, 2002*]) could include within-trial changes in stimulus signal-to-noise ratio (evidence quality) or anticipated reward payout. Task-relevant variability can also arise from internal sources, including noise in neural processing of sensory input and motor output (*Ma and Jazayeri, 2014*; *Faisal et al., 2008*). We assumed subjects do not have precise knowledge of the strength or nature of these noise sources, and thus they could not optimize their strategy accordingly. However, people may be capable of rapidly estimating performance error that results from such internal noise processes and adjusting on-line (*Bonnen et al., 2015*). To extend the models we considered, we could therefore assume that subjects can estimate the magnitude of their own sensory and motor noise, and use this information to adapt their decision strategies to improve performance.

Real subjects likely do not rely on a single strategy when performing a sequence of trials (*Ashwood et al., 2022*) and instead rely on a mix of near-normative, sub-normative, and heuristic strategies. In fitting subject data, experimentalists are thus presented with the difficult task of constructing a library of possible models to use in their analysis. More general approaches have been developed for fitting response data to a broad class of models (*Shinn et al., 2020*), but these model libraries are typically built on pre-existing assumptions of how subjects accumulate evidence and make decisions. Because the potential library of decision strategies is theoretically limitless, a normative analyses can both expand and provide insights into the range of possible subject behaviors in a systematic and principled way. Understanding this scope will assist in developing a well-groomed candidate list of near-normative and heuristic models. For example, if a normative analysis of performance on a dynamic reward task produces threshold dynamics similar to those in *Figure 2B*, then the fitting library should include a piecewise-constant threshold (or urgency signal) model. Combining these model-based investigations with model-free approaches, such as rate-distortion theory (*Berger, 2003*; *Eissa et al., 2021*), can also aid in identifying commonalities in performance and resource usage within and across model classes without the need for pilot experiments.

Our work complements the existing literature on optimal decision thresholds by demonstrating the diversity of forms those thresholds can take under different dynamic task conditions. Several early normative theories were, like ours, based on dynamic programming (*Rapoport and Burkheimer, 1971*; *Busemeyer and Rapoport, 1988*) and in some cases models fit to experimental data (*Ditterich, 2006*). For example, dynamic programming was used to show that certain optimal decisions can require non-constant decision boundaries similar to those of our normative models in dynamic reward tasks (*Frazier and Yu, 2007*; *Figure 2*). More recently, dynamic programming (*Drugowitsch et al., 2012*; *Drugowitsch et al., 2014b*; *Tajima et al., 2016*) or policy iteration (*Malhotra et al., 2017*;

*Malhotra et al., 2018*) have been used to identify normative strategies in dynamic environments that can have monotonically collapsing decision thresholds that in some cases can be implemented using an urgency signal (*Tajima et al., 2019*). These strategies include dynamically changing decision thresholds when signal-to-noise ratios of evidence streams vary according to a Cox-Ingersoll-Ross process (*Drugowitsch et al., 2014a*) and non-monotonic thresholds when the evidence quality varies unpredictably across trials but is fixed within each trial *Malhotra et al., 2018*. Other recent work has started to generalize notions of urgency-gating behavior (*Trueblood et al., 2021*). However, these previous studies tended to focus on environments with a fixed structure, in which dynamic decision thresholds are adapted as the observer acquires knowledge of the environment. Here we have characterized in more detail how both expected and unexpected changes in context within trials relate to changes in decision thresholds over time.

Perceptual decision-making tasks provide a readily accessible route for validating our normative theory, especially considering the ease with which task difficulty can be parameterized to identify parameter ranges in which strategies can best be differentiated (*Philiastides et al., 2006*). There is ample evidence already that people can tune the timescale of leaky evidence accumulation processes to the switching rate of an unpredictably changing state governing the statistics of a visual stimulus, to efficiently integrate observations and make a decision about the state (*Ossmy et al., 2013*; *Glaze et al., 2015*). We thus speculate that adaptive decision rules could be identified similarly in the strategies people use to make decisions about perceptual stimuli in dynamic contexts.

The neural mechanisms responsible for implementing and controlling decision thresholds are not well understood. Recent work has identified several cortical regions that may contribute to threshold formation, such as prefrontal cortex (*Hanks et al., 2015*), dorsal premotor area (*Thura and Cisek, 2020*), and superior colliculus (*Crapse et al., 2018*; *Jun et al., 2021*). Urgency signals are a complementary way of dynamically changing decision thresholds via a commensurate scale in belief, which *Thura and Cisek, 2017* suggest are detectable in recordings from basal ganglia. The normative decision thresholds we derived do not employ urgency signals, but analogous UGMs may involve non-monotonic signals. For example, the switch from an infinite-to-constant decision threshold typical of low-to-high reward switches would correspond to a signal that suppresses responses until a reward change. Measurable signals predicted by our normative models would therefore correspond to zero mean activity during low reward, followed by constant mean activity during high reward. While more experimental work is needed to test this hypothesis, our work has expanded the view of normative and neural decision making as dynamic processes for both deliberation and commitment.

## Materials and methods
### Normative decision thresholds from dynamic programming

Here we detail the dynamic programming tools required to find normative decision thresholds. For the free-response tasks we consider, an observer gathers a sample of evidence $\xi$, uses the log-likelihood ratio (LLR) $y = \frac{\Pr(s_+|\xi)}{\Pr(s_-|\xi)}$ as their 'belief', and sets potentially time-dependent decision thresholds, $\theta_\pm(t)$, that determine when they will stop accumulating evidence and commit to a choice. When $y \geq \theta_+(t)$ ($y \leq \theta_-(t)$), the observer chooses the state $s_+$ ($s_-$). In general, an observer is free to set $\theta_\pm(t)$ any way they wish. However, a normative observer sets these thresholds to optimize an objective function, which we assume throughout this study to be the trial-averaged reward rate, $\rho$, which is given by *Equation 1*. In this definition of reward rate, the incremental cost function $c(t)$ accounts for both explicit costs (e.g. paying for observed evidence, metabolic costs of storing belief in working memory) and implicit costs (e.g. opportunity cost). We assume symmetry in the problem (in terms of prior, rewards, etc.) that guarantees the thresholds are symmetric about $y = 0$ and $\theta_\pm(t) = \pm\theta(t)$. We derive the optimal threshold policy for a general incremental cost function $c(t)$, but in our results we consider only constant costs functions $c$. Although the space of possible cost functions is large, restricting to a constant value ensures that threshold dynamics are governed purely by task and reward structure and not by an arbitrary evidence cost function.

To find the thresholds $\pm\theta(t)$ that optimize the reward rate given by *Equation 1*, we start with a discrete-time task where observations are made every $\delta t$ time units, and we simplify the problem so the length of each trial is fixed and independent of the decision time $T_d$. This simplification makes the denominator of $\rho$ constant with respect to trial-to-trial variability, meaning we can optimize reward

rate by maximizing the numerator $\langle R \rangle - \langle C(T_d) \rangle$. Under this simplified task structure, we suppose the observer has just drawn a sample $\xi_n$ and updated their state likelihood to $p_n = \frac{1}{1+e^{-y_n}}$, where $y_n = \ln \frac{\Pr(s_+|\xi_{1:n})}{\Pr(s_-|\xi_{1:n})}$ is the discrete-time LLR given by **Equation 2**. At this moment, the observer takes one of three possible actions:

1. Stop accumulating evidence and commit to choice $s_+$. This action has value equal to the average reward for choosing $s_+$, which is given by:

$$V_+(p_n) = R_c p_n + R_i(1 - p_n), \tag{9}$$

   where $R_c$ is the value for a correct choice and $R_i$ is the value for an incorrect choice.

2. Stop accumulating evidence and commit to choice $s_-$. By assuming the reward for correctly (or incorrectly) choosing $s_+$ is the same as choosing $s_-$, the value of this action is obtained by symmetry from:

$$V_-(p_n) = R_c(1 - p_n) + R_i p_n. \tag{10}$$

3. Wait to commit to a choice and draw an additional piece of evidence. Choosing this action means the observer expects their future overall value $V$ to be greater than their current value, less the cost incurred by waiting for additional evidence. Therefore, the value of this choice is given by:

$$V_w(p_n) = \langle V(p_{n+1})|p_n \rangle_{p_{n+1}} - c(t)\delta t, \tag{11}$$

   where $c$ is the incremental evidence cost function; because we assume that the incremental cost is constant, this simplifies $c(t)\delta t = c\delta t$.

Given the action values from **Equations 9–11**, the observer takes the action with maximal value, resulting in their overall value function

$$
\begin{aligned}
V(p_n) \quad &= \max\{V_+(p_n), V_-(p_n), V_w(p_n)\} \\
&= \max \begin{cases} R_c p_n + R_i(1 - p_n) & \text{choose } s_+ \\ R_c(1 - p_n) + R_i p_n & \text{choose } s_- \\ \langle V(p_{n+1})|p_n \rangle_{p_{n+1}} - c\delta t, & \text{sample again} \end{cases}
\end{aligned} \tag{12}
$$

Because the value-maximizing action depends on the state likelihood, $p_n$, the regions of likelihood space where each action is optimal divide the space into three disjoint regions. The boundaries of these regions are exactly the optimal decision thresholds, which can be mapped to LLR-space to obtain $\pm\theta(t)$. To find these thresholds numerically, we started by discretizing the state likelihood space $p_n$. Because the state likelihood $p_n$ is restricted to values between 0 and 1, whereas the log-likelihood ratio $y_n = \ln \frac{p_n}{1-p_n}$ is unbounded, we chose to formulate all the components of Bellman's equation in terms of $p_n$ to minimize truncation errors. We then proceeded by using backward induction in time, starting at the total trial length $t = T_t$. At this moment in time, it impossible to wait for more evidence, so the value function in **Equation 12** does not depend on the future. This approach implies that the value function is:

$$
\begin{aligned}
V(p_n) \quad &= \max\{V_+(p_n), V_-(p_n), V_w(p_n)\} \\
&= \max \begin{cases} R_c p_n + R_i(1 - p_n) & \text{choose } s_+ \\ R_c(1 - p_n) + R_i p_n & \text{choose } s_- \end{cases}
\end{aligned}
$$

Once the value is calculated at this time point, it can be used as the future value at time point $t = T_t - \delta t$.

To find the decision thresholds for the desired tasks where $T_t$ is not fixed, we must optimize both the numerator and denominator of **Equation 1**. To account for the variable trial length, we adopt techniques from average reward reinforcement learning (**Mahadevan, 1996**) and penalize the waiting time associated with each action by the waiting time itself scaled by the reward rate $\rho$ (i.e., $\langle t_i \rangle \rho$ for committing to $s_+$ or $s_-$ and $\rho\delta t$ for waiting). This modification makes all trials effectively the same length and allows us to use the same approach used to derive **Equation 12** (**Drugowitsch et al., 2012**). The new overall value function is given by **Equation 3**:

$$V(p_n; \rho) \quad = \max\{V_+(p_n; \rho), V_-(p_n; \rho), V_w(p_n; \rho)\}$$

$$= \max \begin{cases} R_c p_n + R_i(1-p_n) - \langle t_i \rangle \rho, & \text{choose } s_+ \\ R_c(1-p_n) + R_i p_n - \langle t_i \rangle \rho, & \text{choose } s_- \\ \langle V(p_{n+1}; \rho) | p_n \rangle_{p_{n+1}} - c(t)\delta t - \rho \delta t, & \text{sample again} \end{cases} \qquad (13)$$

To use this new value function to numerically find the decision thresholds, we must note two new complications that arise from moving away from fixed-length trials. First, we no longer have a natural end time from which to start backward induction. We remedy this issue by following the approach of **Drugowitsch et al., 2012** and artificially setting a final trial time $T_f$ that is far enough in the future so that decision times of this length are highly unlikely and do not impact the response distributions. If we desire accurate thresholds up to a time $t$, we set $T_f = 5t$, which produces an accurate solution while avoiding a large numerical overhead incurred from a longer simulation time. In our simulations, we set $t$ based on when we expect most decisions to be made. Second, the value function now depends on the unknown quantity $\rho$, resulting in a co-optimization problem. To address this complication, note that when $\rho$ is maximized, our derivation requires $V\left(p_0 = \frac{1}{2}; \rho\right) = 0$ for a consistent Bellman's equation (**Drugowitsch et al., 2012**). We exploit this consistency requirement by fixing an initial reward rate $\rho_0$, solving the value function through backward induction, calculating $V(0; \rho_0)$, and updating the value of $\rho$ via a root finding scheme. For more details on numerical implementation, see https://github.com/nwbarendregt/AdaptNormThresh; **Thresh, 2022**.

## Dynamic context 2AFC tasks

For all dynamic context tasks, we assume that observations follow a Gaussian distribution with so that $\xi|s_\pm \sim \mathcal{N}(\pm\mu, \sigma^2)$. Using the Functional Central Limit Theorem, one can show (**Bogacz et al., 2006**) that in the continuous-time limit, the belief $y$ evolves according to a stochastic differential equation:

$$dy = \pm m \, dt + \sqrt{2m} \, dW_t. \qquad (14)$$

In **Equation 14**, $m = \frac{2\mu^2}{\sigma^2}$ is the scaled signal-to-noise ratio (SNR) given by the observation distribution function $\xi|s_\pm \sim \mathcal{N}(\pm\mu, \sigma^2)$, $dW_t$ is a standard increment of a Wiener process, and the sign of the drift $\pm m \, dt$ is given by the sign of the correct choice $s_\pm$. To construct Bellman's equation for this task, we start by discretizing time $t_{1:n}$ and determine the average value gained by waiting and collecting another observation given by **Equation 4**:

$$\langle V(p_{n+1}; \rho) | p_n \rangle_{p_{n+1}} = \int_0^1 V(p_{n+1}; \rho) f_P(p_{n+1} | p_n) \, dp_{n+1},$$

where $p_n = \Pr(s_+|\xi_{1:n})$ is the probability the environment is in state $s_+$ given $n$ pieces of evidence. The main difficulty in computing this expectation is computing the conditional probability distribution $f_P(p_{n+1} | p_n)$, which we call the likelihood transfer function. Once we construct the likelihood transfer function, we can use our discretization of the state likelihood space $p_n$ to evaluate the integral in **Equation 4** using any standard numerical quadrature scheme. To compute this transfer function, we can start by using the definition of the LLR $y_n$ and leveraging the relationship between $p_n$ and $y_n$ to find $p_n$ and a function of the observation $\xi_n$:

$$\begin{aligned} p_{n+1} \quad &= \frac{1}{1+e^{-y_{n+1}}} = \frac{1}{1+e^{-\ln\frac{f_+(\xi_{n+1})}{f_-(\xi_{n+1})}e^{-y_n}}} \\ &= \frac{1}{1+\frac{f_-(\xi_{n+1})}{f_-(\xi_{n+1})}\left(\frac{1-p_n}{p_n}\right)} = \frac{p_n}{p_n + (1-p_n)e^{-\frac{2\xi_{n+1}\mu}{\sigma^2}}}. \end{aligned} \qquad (15)$$

Note that we used the fact that in discrete-time with a time step $\delta t$, the observations $\xi|s_\pm \sim \mathcal{N}(\pm\mu\delta t, \sigma^2\delta t)$. The relationship between $\xi_{n+1}$ and $p_{n+1}$ in **Equation 15** can be inverted to obtain:

$$\xi_{n+1} = \frac{\sigma^2}{2\mu} \ln \frac{(p_n-1)p_{n+1}}{p_n(p_{n+1}-1)}.$$

With this relationship established, we can find the likelihood transfer function $f_p(p(\xi_{1:n+1})|p(\xi_{1:n}))$ by finding the observation transfer function $f_\xi(\xi(p_{n+1})|\xi(p_n))$ and performing a change of variables, which by independence of the sample is simply $f_\xi(\xi_{n+1})$. With probability $p_n$, $\xi_{n+1}$ will be drawn from the normal distribution $\mathcal{N}(+\mu\delta t, \sigma^2\delta t)$, and with probability $1 - p_n$, $\xi_{n+1}$ will be drawn from the normal distribution $\mathcal{N}(-\mu\delta t, \sigma^2\delta t)$. This immediately provides the observation transfer function by marginalizing:

$$f_\xi(\xi_{n+1}|\xi_{1:n}) = p_n \left\{ \frac{1}{\sqrt{2\pi\delta t}\sigma} e^{-\frac{(\xi_{n+1}-\mu\delta t)^2}{2\sigma^2\delta t}} \right\} + (1 - p_n) \left\{ \frac{1}{\sqrt{2\pi\delta t}\sigma} e^{-\frac{(\xi_{n+1}+\mu\delta t)^2}{2\sigma^2\delta t}} \right\}.$$

Performing the change of variables using the derivative $\frac{d\xi_{n+1}}{dp_{n+1}} = \frac{\sigma^2}{2p_{n+1}\mu - 2p_{n+1}^2\mu} > 0$ yields the transfer function

$$f_p(p_{n+1}|p_n) = \frac{1}{2\mu p_{n+1}(1 - p_{n+1})\sqrt{2\pi\delta t}\sigma} \left[ p_n \exp\left\{ -\frac{1}{2\sigma^2\delta t} \left( \frac{\sigma^2}{2\mu}\ln\frac{(p_n - 1)p_{n+1}}{p_n(p_{n+1} - 1)} - \delta t\mu \right)^2 \right\} \right.$$
$$\left. + (1 - p_n)\exp\left\{ -\frac{1}{2\sigma^2\delta t} \left( \frac{\sigma^2}{2\mu}\ln\frac{(p_n - 1)p_{n+1}}{p_n(p_{n+1} - 1)} + \delta t\mu \right)^2 \right\} \right].$$

(16)

Note that **Equation 16** is equivalent to the likelihood transfer function given by Equation 16 in **Drugowitsch et al., 2012** for the case of $m = 1$. Combining **Equation 14** and **Equation 16**, we can construct Bellman's equation for any dynamic context task.

### Reward-change task thresholds

For the reward-change task, we fixed punishment $R_i = 0$ and allowed the reward $R_c$ to be a Heaviside function given by **Equation 5**:

$$R_c(t) = (R_2 - R_1)H_\theta(t - 0.5) + R_1.$$

In **Equation 5**, there is a single switch in rewards between pre-change reward $R_1$ and post-change reward $R_2$. This change occurs at $t = 0.5$. Substituting this reward function into **Equation 3** allows us to find the normative thresholds for this task as a function of $R_1$ and $R_2$.

For the inferred reward change task, we allowed the reward value $R(t) \in \{R_H, R_L\}$ to be controlled by a continuous-time two-state Markov process with transition (hazard) rate $h$ between rewards $R_H \geq R_L$. The hazard rate $h$ governs the probability of switching between $R_H$ and $R_L$:

$$\Pr(R(t + \delta t) = R_{H/L}|R(t) = R_{L/H}) = h\delta t + o(\delta t), \; \delta t \downarrow 0,$$
$$\Pr(R(t + \delta t) = R_{H/L}|R(t) = R_{H/L}) = 1 - h\delta t + o(\delta t), \; \delta t \downarrow 0,$$

where $o(\delta t)$ represents a function $g(\delta t)$ with the property $\lim_{\delta t \downarrow 0} \frac{g(\delta t)}{\delta t} = 0$ (i.e., all other terms are of smaller order than $\delta t$). In addition, the state of this Markov process must be inferred from evidence $\eta$ that is independent of the environment's state evidence $\xi$ (i.e., the correct choice). For simplicity, we assume that the reward-evidence source is also Gaussian-distributed such that $\eta|R_{H/L} \sim \mathcal{N}(\pm\mu_R, \sigma_R^2)$ with quality $m_R = \frac{2\mu_R^2}{\sigma_R^2}$. **Glaze et al., 2015**; **Veliz-Cuba et al., 2016**; **Barendregt et al., 2019** have shown that the belief $y_R = \ln\frac{\Pr(R(t)=R_H|\eta)}{\Pr(R(t)=R_L|\eta)}$ for such a dynamic state inference process is given by the modified DDM

$$dy_R = x(t)m_R\,dt - 2h\sinh(y_R)\,dt + \sqrt{2m_R}\,dW_t,$$

where $x(t) \in \pm 1$ is a telegraph process that mirrors the state of the reward process (i.e., $x(t) = 1$ when $R(t) = R_H$ and $x(t) = -1$ when $R(t) = R_L$). With this belief over reward state, we must also modify the values $V_+(p_n; \rho)$ and $V_-(p_n; \rho)$ to account for the uncertainty in $R_c$. Defining $q = \frac{e^{y_R}}{1+e^{y_R}}$ as the reward likelihood gives

$$V_+(p_n; \rho) = (R_H q_n + R_L(1 - q_n))p_n - \langle t_i \rangle \rho,$$
$$V_-(p_n; \rho) = (R_H q_n + R_L(1 - q_n))(1 - p_n) - \langle t_i \rangle \rho,$$

where we have fixed $R_i = 0$ for simplicity.

### SNR-change task thresholds

For the SNR-change task, we allowed the task difficulty $m = \frac{2\mu^2}{\sigma^2}$ to vary over a single trial by making $\mu(t)$ a time-dependent step function given by **Equation 6**:

$$\mu(t) = (\mu_2 - \mu_1)H_\theta(t - 0.5) + \mu_1.$$

In **Equation 6**, there is a single switch in evidence quality between pre-change quality $\mu_1$ and post-change quality $\mu_2$. This change occurs at $t = 0.5$. Substituting this quality time series into the likelihood transfer function in **Equation 16** allows us to find the normative thresholds for this task as a function of $\mu_1$ and $\mu_2$. This modification necessitates that the transfer function $f_p$ also be a function of time; however, because the quality change points are known in advance to the observer, we can simply change between different transfer functions at the specified quality changes.

## Reward-change task model performance

Here we detail the three models used to compare observer performance in the reward-change task, as well as the noise filtering process used to generate synthetic data. For the noisy Bayesian model, the observer uses the thresholds $\pm\theta(t)$ obtained via dynamic programming, thus making the observer a noisy ideal observer. For the constant-threshold model, the observer uses a constant threshold $\pm\theta(t) = \pm\theta_0$, which is predicted to be optimal only in very simple, static decision environments with only two states $s$. Both the noisy Bayesian and constant-threshold models also use a noisy perturbation of the LLR $\tilde{y} = y + \sigma_y Z$ as their belief, where $\sigma_y$ is the strength of the noise and $Z$ is a sample from a standard normal distribution. In continuous-time, this perturbation involves adding an independent Wiener process to **Equation 14**:

$$d\tilde{y} = \pm m\,dt + \sqrt{2m}\,dW_t + \sigma_y\,dW_t',$$

where $dW_t'$ is an independent Wiener process with strength $\sigma_y$. The UGM, being a phenomenological model, behaves differently from the other models. The UGM belief $E$ is the output of the noisy low-pass filter given by **Equation 7**:

$$\tau\,dE = \left( -E + \frac{1}{1 + e^{-y}} - \frac{1}{2} \right)\,dt + \sigma_y\,dW_t.$$

To add additional noise to the UGM's belief variable $E$, we simply allowed $\sigma_y > 0$ in the low-pass filter in **Equation 7**.

In addition to the inference noise with strength $\sigma_y$, we also filtered each process through a Gaussian response-time filter with zero mean and standard deviation $\sigma_{mn}$. Under this response-time filter, if the model predicted a response time $T$, the measured response time $\tilde{T}$ was drawn from a normal distribution centered at $T$ with standard deviation $\sigma_{mn}$. If the response time $\tilde{T}$ was drawn outside of the simulation's time discretization (i.e., if $\tilde{T} < 0$ or $\tilde{T} > \frac{T_f}{5}$), we redrew $\tilde{T}$ until it fell within the discretization. This filter was chosen to represent both "early responses" caused by attentional lapses, as well as 'late responses' caused by motor processing delays between the formation of a choice in the brain and the physical response. We have chosen to add these two sources of noise after optimizing each model to maximize average reward rate, rather than reoptimizing each model after adding these additional noise sources. Although we could have reoptimized each model to maximize performance across noise realizations, we were interested in how the models responded to perturbations that drove their performance to be sub-optimal (but possibly near-optimal).

To compare model performance on the reward-change task, we first fixed the value of pre-change reward $R_1$ (and set $R_1 + R_2 = 11$) to find the post-change reward and tuned each model to achieve optimal reward rate with no additional noise in both the inference and response processes. Bellman's equation outputs both the optimal normative thresholds and reward rate. For the constant threshold model and the UGM, we approximated the maximal performance of each model by using a grid search over each models parameters to find the model tuning that yielded the highest average reward rate. After tuning all models for a given reward structure, we filtered them through both the sensory ($\sigma_y$) and motor ($\sigma_{mn}$) noise sources without re-turning the models to account for this additional noise. When generating noisy synthetic data from these models, we generated 100 synthetic subjects, each

with sampled values of $\sigma_y$ and $\sigma_{mn}$. For each synthetic subject with noise parameter sample $(\sigma_y, \sigma_{mn})$, we defined the "noise strength" of that subject's noise to be the ratio

$$\frac{\sigma_y + \sigma_{mn}}{\overline{\sigma}_y + \overline{\sigma}_{mn}},$$

where $\overline{\sigma}_y = 5$ and $\overline{\sigma}_{mn} = 0.25$ are the maximum values of belief noise and motor noise considered, respectively. Using this metric, noise strength is defined between 0 and 1. Additionally, the maximum noise levels $\overline{\sigma}_y$ and $\overline{\sigma}_{mn}$ where chosen such that a noise strength of 0.5 is approximately equivalent to the fitted noise strength obtained from tokens task subject data. We plot the response distributions using noise strengths of 0, 0.5, and 1 in our results. To compare the performance of each model after being corrupted by noise, we then generated 1000 trials for each subject and had each simulated subject repeat the same block of trials three times, one for each model. This process ensured that the only difference between model performance would come from their distinct threshold behaviors, because each model was taken to be equally noisy and was run using the same stimuli.

## Tokens task

### Normative model for the tokens task

For the tokens task, observations in the form of token movements are Bernoulli distributed with parameter $p = 0.5$ that occur every 200ms. Once a subject committed to a decision, the token movements continued at a faster rate until the entire animation had finished. This post-decision token acceleration was 170ms per movement in the 'slow' version of the task and 20ms per movement for the 'fast' version of the task. Because of the stimulus structure, one can show using a combinatorial argument (*Cisek et al., 2009*) that the likelihood function $p_n$ is given by *Equation 8*. Constructing the likelihood transfer function $f_p$ required for Bellman's equation is also simplified from the Gaussian 2AFC tasks, as there are only two possible likelihoods that one can transition two after observing a token movement:

$$f_p(p(\text{top}\,|U_{n+1},L_{n+1},C_{n+1})|p(\text{top}\,|U_n,L_n,C_n)) = \begin{cases} \frac{1}{2}, & (U_{n+1},L_{n+1},C_{n+1}) = (U_n+1,L_n,C_n-1) \\ \frac{1}{2}, & (U_{n+1},L_{n+1},C_{n+1}) = (U_n,L_n+1,C_n-1) \\ 0, & \text{otherwise} \end{cases} \quad (17)$$

Combining *Equation 8* and *Equation 17*, we can fully construct Bellman's equation for the tokens task. While the timings of the token movements, post-decision token acceleration, and inter-trial interval are fixed, we let the reward $R_c$ and cost function $c$ be free parameters to control the different threshold dynamics of the model.

### Model fitting and comparison

We used three models to fit the subject response data provided by *Cisek et al., 2009*: the noisy Bayesian model ($k = 4$ parameters), the constant threshold model ($k = 3$ parameters), and the UGM ($k = 5$) parameters (*Table 1*). To adapt the continuous-time models to this discrete-time task, we simply changed the time step to match the time between token movements ($\delta t = 200$ ms). To fit each model, we took the subject response time distributions as our objective function and used Markov Chain Monte Carlo (MCMC) with a standard Gaussian proposal distribution to generate an approximate posterior made up of 10,000 samples. For more details as to our specific implementation of MCMC for this data, see the MATLAB code available at https://github.com/nwbarendregt/Adapt-NormThresh, (copy archived at swh:1:rev:2878a3d9f5a3b9b89a0084a897bef3414e9de4a2; *Thresh, 2022*). We held out 2 of the 22 subjects to use as training data when tuning the covariance matrix of the proposal distribution for each model, and performed the model fitting and comparison analysis

**Table 1.** List of model parameters used for analyzing tokens task response time data.

| NB: | | Const: | | UGM: | |
|---|---|---|---|---|---|
| | Reward $R_c$ | | Threshold $\theta_0$ | | Threshold Scale $\theta_0$ |
| | Cost $c(t) = c$ | | Sensory Noise $\sigma_y$ | | Gain $a$ |
| | Sensory Noise $\sigma_y$ | | Motor Noise $\sigma_{mn}$ | | Sensory Noise $\sigma_y$ |
| | Motor Noise $\sigma_{mn}$ | | | | Time Constant $\tau$ |
| | | | | | Motor Noise $\sigma_{mn}$ |

on the remaining 20 subjects. Using the approximate posterior obtained via MCMC for each subject and model, we used calculated AICc using the formula

$$\text{AICc} = 2k - 2\ln\left(\hat{L}\right) + \frac{2k^2 + 2k}{n - k - 1}. \tag{18}$$

In **Equation 18**, $k$ is the number of parameters of the model, $\hat{L}$ is the likelihood of the model evaluated at the maximum-likelihood parameters, and $n$ is the number of responses in the subject data (**Cavanaugh, 1997**; **Brunham and Anderson, 2002**). Because each subject performed different numbers of trials, using AICc allowed us to normalize results to account for the different data sizes; note that for many responses (i.e., for large $n$), AICc converges to the standard definition of AIC. For the second model selection metric, we measured how well each fitted model predicted the trial-by-trial responses of the data by calculating the average RMSE between the response times from the data and the response times predicted by each model. To measure the difference between a subject's response time distribution and the fitted model's distribution (**Figure 6—figure supplement 1**), we used Kullback-Leibler (KL) divergence:

$$\text{KL} = \sum_{i=0}^{15} \text{RT}_D(i) \ln\left(\frac{\text{RT}_D(i)}{\text{RT}_M(i)}\right). \tag{19}$$

In **Equation 19**, $i$ is a time index representing the number of observed token movements, $\text{RT}_D(i)$ is the probability of responding after $i$ token movements from the subject data, and $\text{RT}_M(i)$ is the probability of responding after $i$ token movements from the model's response distribution. Smaller values of KL divergence indicate that the model's response distribution is more similar to the subject data.

## Code availability

See https://github.com/nwbarendregt/AdaptNormThresh; (copy archived at swh:1:rev:2878a3d-9f5a3b9b89a0084a897bef3414e9de4a2; **Thresh, 2022**) for the MATLAB code used to generate all results and figures.

## Acknowledgements

We thank Paul Cisek for providing response data from the tokens task used in our analysis.

## Additional information

### Competing interests

Joshua I Gold: Senior editor, *eLife*. The other authors declare that no competing interests exist.

### Funding

| Funder | Grant reference number | Author |
| --- | --- | --- |
| National Institutes of Health | R01-MH-115557 | Nicholas W Barendregt<br>Joshua I Gold<br>Krešimir Josić<br>Zachary P Kilpatrick |
| National Institutes of Health | R01-EB029847-01 | Nicholas W Barendregt<br>Zachary P Kilpatrick |
| National Science Foundation | NSF-DMS-1853630 | Nicholas W Barendregt<br>Zachary P Kilpatrick |
| National Science Foundation | NSF-DBI-1707400 | Krešimir Josić |

The funders had no role in study design, data collection and interpretation, or the decision to submit the work for publication.

## Author contributions
Nicholas W Barendregt, Conceptualization, Software, Formal analysis, Investigation, Visualization, Methodology, Writing - original draft, Writing - review and editing; Joshua I Gold, Krešimir Josić, Zachary P Kilpatrick, Supervision, Funding acquisition, Writing - review and editing

## Author ORCIDs
Nicholas W Barendregt http://orcid.org/0000-0002-3268-9426
Joshua I Gold http://orcid.org/0000-0002-6018-0483
Krešimir Josić http://orcid.org/0000-0002-1975-3913
Zachary P Kilpatrick http://orcid.org/0000-0002-2835-9416

## Decision letter and Author response
Decision letter https://doi.org/10.7554/eLife.79824.sa1
Author response https://doi.org/10.7554/eLife.79824.sa2

## Additional files

### Supplementary files
• MDAR checklist

### Data availability
MATLAB code used to generate all results and figures is available at https://github.com/nwbarendregt/AdaptNormThresh, (copy archived at swh:1:rev:2878a3d9f5a3b9b89a0084a897bef3414e9de4a2).

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
