## [Editor Report]

This paper makes an important contribution to the study of decision-making under time pressure. The authors provide convincing evidence that decision boundaries can be highly nontrivial – even reaching infinity in realistic regimes. This paper will be of broad interest to both experimentalists and theorists working on decision-making under time pressure.

---

## [Decision Letter]

**Decision letter after peer review:**

Thank you for submitting your article "Normative Decision Rules in Changing Environments" for consideration by *eLife*. Your article has been reviewed by 3 peer reviewers, one of whom is a member of our Board of Reviewing Editors, and the evaluation has been overseen Timothy Behrens as the Senior Editor. The following individual involved in the review of your submission has agreed to reveal their identity: Gaurav Malhotra (Reviewer #3).

All three reviewers very much liked the paper. It was nice to see the formalism used to solve these problems take a central place in the manuscript, and the huge variability in bounds is something we haven't seen before.

But that didn't stop us from making a huge number of comments – you have either the good luck or the bad luck, depending on your point of view, of being reviewed by experts. Most comments have to do with the presentation: important information was missing (or at least we couldn't find it), and there were even places where we got lost (not a good sign, given that all three of us work in the field). Details follow.

I. The tokens task can be analyzed using the formalism introduced in Equation (3), but it seems pretty far from the "dynamic context" examples emphasized in the bulk of the paper. That doesn't mean the tokens task shouldn't be included. But it does mean we have no evidence one way or the other whether subjects would adopt the highly idiosyncratic boundaries found in simulations (for instance, the infinite threshold boundaries in Figures 2i, 2ii).

You need to be clear about this. The way the paper reads, it sounds like you have provided evidence for the dynamic context setup, when in fact that's not the case. It should be crystal clear that dynamic context problems have not been explored, at least not in the lab. Instead, what you showed is that a normative model can beat a particular heuristic model.

II. The following are technical but important.

1. Adding noise: you are currently optimizing the model parameters/policy before adding sensory and motor noise. Decision-makers could be aware of sensory noise and so could try to optimize their decision processes with that knowledge. Would you be able to also compare the models' performances if they have been optimized to maximize performance in the presence of all noise? From our understanding, this should be feasible for the Const and UGM model, but might be harder for the normative model. Sensory noise could be included by finding Equation (13) that includes such noise, but finding the optimal thresholds once RT noise is included might be prohibitive. This is just a suggestion, not an essential inclusion. However, it might be worth at least discussing the difference between what you do, and being clear on the scenario you considered.

2. Fitting token task data: according to Cisek et al. (2009), the same participants performed both the slow and the fast version of the task. However, their fitted reward magnitudes differ by an order of magnitude between the two conditions (your Figure 6C/F). Is it just that the fitting objective didn't well-constrain these parameters? Given that you use MCMC for model fits, you could compare the parameter posteriors across conditions. Furthermore, how much worse would the model fits become if you would fit both conditions simultaneously and share all parameters that can be meaningfully shared across conditions? In any case, an explanation for this difference should be provided in the manuscript.

3. The dynamic context examples would seem to apply only when subjects take many seconds to make a decision. This would seem to rule out perceptual decision-making tasks. Is this true? If so, you should be upfront about this – so that those who work on perceptual decision-making will know what they're getting into.

4. A known and predictable change in the middle of a task seems somewhat unrealistic. Given that it plays such a central role, concrete examples where this comes up would be very helpful. Or at least you should make a proposal for laboratory experiments where it could come up. The examples in the introduction ("Some of these factors can change quickly and affect our deliberations in real time; e.g., an unexpected shower will send us hurrying down the faster route (Figure 1A), whereas spotting a new ice cream store can make the longer route more attractive.") don't quite fall into the "known and predictable change" category.

III. Better contact with existing literature needs to be made. For instance:

1. Drugowitsch, Moreno-Bote and Pouget (2014) already computed normative decision policies for time-varying SNR, with the difference that they assumed the SNR to follow a stochastic process rather than a known, deterministic time course. Thus, the work is closely related, but not equivalent.

2. Some early models to predict dynamic decision boundaries were proposed by Busemeyer and Rapoport (1988) and Rapoport and Burkheimer (1971) in the context of a deferred decision-making task.

3. One of the earliest models to use dynamic programming to predict non-constant decision boundaries was Frazier and Yu (2007). Indeed some boundaries predicted by the authors (e.g. Figure 2v) are very similar to boundaries predicted by this model. In fact, the switch from high to low reward used to propose boundaries in Figure 2v can be seen as a "softer" version of the deadline task in Frazier and Yu (2007).

4. Another early observation that time-varying boundaries can account for empirical data was made by Ditterich (2006). Seems highly relevant to the authors' predictions, but is not cited.

5. The authors seem to imply that their results are the first results showing non-monotonic thresholds. This is not true. See, for example, Malhotra et al. (2018). What is novel here is the specific shape of these non-monotonic boundaries.

IV. Clarity could be massively improved. If you want to write an unclear paper that is your prerogative. However, if you do, you can't say "Our results can aid experimentalists investigating the nuances of complex decision-making in several ways". It would be difficult to aid experimentalists if they have to struggle to understand the paper.

Below are comments collected from the three reviewers, and more or less collated (so it's possible there's some overlap, and the order isn't exactly optimized). You can, in fact, almost ignore them, if you take into account the main message: all information should be easily accessible, in the main text, and the figures should be easy to make sense of.

As authors, we are aware that the length of replies can sometimes exceed the paper, which is not a good use of anybody's time. Please use your judgment as to which ones you reply to. For instance, if you're going to implement our suggestions, no reason to tell us. Maybe comment only if you have a major objection? Or use some other scheme? What we really care about is that the revised paper is easy to read!

1. When the UGM was introduced, all you say is "urgency-gating models (UGMs) use thresholds that collapse monotonically over time". You include some references, but for the casual reader, it looks like you're considering a generic collapsing bound model. In fact, you're considering a particular shape for the collapsing bound and particular filtering of the evidence. This should be clear. It also needs to be justified. For instance, Voskuilen et al. (J. Math. Psych. 73:59-79, 2016) use a different functional form for the collapsing bound, and they don't filter the evidence. Why use one model over another?

And while we're on the topic of the UGM: Equation (4) low-pass filters the noise-free observer's belief y that reflects all accumulated evidence up to current time t. According to our reading of Cisek et al. (2009), the UGM low-pass filters the momentary internal estimate of sensory information (the Ei(tau) defined below Equation (1); Equations. (17)-(19) for the low-pass filter in Cisek et al.) rather than the accumulated estimate of sensory information. Are we misinterpreting Cisek et al. (2009) or your Equation. (4)? Either way, please clarify.

In Equation. 4 it would be more clear to put -E + 0.5*tanh(y) on the RHS. What's the justification for tanh? Why not just filter y? Do you use tanh because the original paper did? If so, you should point that out.

Also, what's y in that equation?

2. Important inline equations need to be displayed. There's nothing more annoying than having to crawl through text to look for the definition of an important symbol. To take a few (hardly exhaustive) examples: f±(ξ),y,pn,fp(pn+1|pn). The actual list is much longer. If any symbol is going to be used again, please make it easy to find! This in itself is a reason for displayed equations: you can refer to equation numbers when introducing variables that you haven't used for several pages.

3. A lot of the lines don't have line numbers, which is relevant mainly for us, since it's hard to refer to things without line numbers. This is a bug, but there's a way to fix it. I think (but I'm not sure) that in your latex file you need to leave a space between equations and surrounding text. (Or maybe no space? It's been a while). Although I believe there's a more elegant fix.

4. Not all equations were numbered. We know, in some conventions only equations one refers to are numbered (that's what one of us grew up with), but it turns out to be not so convenient for us as reviewers when we want to refer to an un-numbered equation.

5. Lines 43-6: "Efforts to model decision-making thresholds under dynamic conditions have focused largely on heuristic strategies. For instance, "urgency-gating models" (UGMs) use thresholds that collapse monotonically over time (equivalent to dilating the belief in time) to explain decisions based on time-varying evidence quality".

In fact, a collapsing bound is not necessarily a heuristic; it can be optimal, although the exact shape of the collapsing bound has to be found by dynamic programming. Please reword to reflect this.

6. Line 76: c(t) is barely motivated at all here. It's better motivated in Methods, but its value is very hard to justify. Why not stick with optimizing average reward, for which c=0? And I don't think you ever told us what c(t) was; only that it was constant (although we could have missed its value).

7. Figure 2C would be easier to make sense of if it were square.

8. In general, information is scattered all over the place, and much of it seems to be missing. Each task should be described succinctly in the main text, with enough information to actually figure out what's going on. In addition, there should be a table listing _all_ the parameters; right now the reader has to go to Methods, and even then it seems that many are missing. For instance, we don't think we were ever told the value of tau in Equation. 4.

9. Lots of questions/comments about Figure 4:

a. It would be very helpful to include the optimal model. I think NB is the optimal model when σ_y=0, but I also believe that in most panels σ_y \ne 0.

b. It would be helpful to emphasize, in the figure caption, that NB with σ_y = 0 is the optimal model. Assuming that's true.

c. Figure 4A: What's the post-reward rate? And please indicate the pre-reward rate at which pre-reward = post-reward. Also, If pre and post-reward rates sum to 11 (as mentioned in Methods, line 411), why are the curves' minima at around 5 rather than 5.5?

g. Figure 4B: horizontal axis label missing (presumably "pre reward"?). And we assume you used the following color code: Orange: reward(NB)-reward(Const); violet: reward(NB)-reward(UGM). Correct? Either way, this should be stated in the figure caption.

e. Figure 4C: what are the pre and post-rewards? And presumably noise strength = σ_y? This should be stated clearly. And more explanation, in the main text, of what "noise strength" is would help.

f. Figure 4F: It is not clear to us why UGM in 0 noise condition have RTs aligned to the time reward increases from R1 to R2. Surely, this model does not take RR into account to compute the thresholds, does it? In fact, looking at Figure 4B, Supplement 1, the thresholds are always highest at t=0. Please clarify.

10. Lines 207-9: "Because the total number of tokens was finite and known to the subject, token movements varied in their informativeness within a trial, yielding a dynamic and history-dependent evidence quality that, in principle, could benefit from adaptive decision processes".

To us, "history-dependent" implies non-Markov, whereas the tokens task is Markov. But maybe that's not what history-dependent means here? This should be clarified.

11. We assume the y-axis in Figure 5i-iv is the difference between the number of tokens on the top and the number on the bottom. This should be stated (if it's true). And please explain how you differentiate between motifs iii and iv. We believe it's the presence of two threshold increases (rather than just one) in motif iv, but we're not sure.

12. What's the reward/punishment structure for the tokens task? It seemed that this was only half explained.

13. Lines 229-232: "To determine the relevance of these adaptive decision strategies to human behavior, we fit discrete-time versions of the noisy Bayesian (four free parameters), constant-threshold (three free parameters), and urgency-gating (five free parameters) models to response-time data from the tokens task collected by Cisek et al. (2009)."

As mentioned above, the parameters should go in a table.

14. You should tell us what V(T_final) is, and why. We believe it's the same as V(0), but We could be wrong.

15. After Equation. 11: it says m = 2 mu^2^/σ^2^. Are these mu and σ different than the ones on line 383? If so, that should be clear. (If not, we're lost.)

16. We looked, but couldn't find, the definition of f_p. We believe it's just a conditional probability,

f_p(p_{n+1}|p_n) = P(p_{n+1}|p_n).

If so, why not use that notation? It would be a lot easier to remember. In any case, when this is used, please tell us what it is, or where it was originally defined (which should be in a displayed equation!).

17. State space is parameterized by p_n, and that needs to be discretized, right? If so, that's worth mentioning. If not, we're lost.

18. Analysis (in particular Equation. 13) would be a lot easier if you used y_n instead of p_n. y_n is what is generally accumulated in DDMs, and it's what you generally plot on the y-axis. So why use p_n?

19. Equations. 14 and 15 should really be in the main text. They're simple and important.

20. We didn't understand the inferred reward change task, in the text starting after line 393. We might have been able to guess, but please put in equations so it's crystal clear.

21. Somewhere below line 404: "a constant threshold … is predicted to be optimal only in simple, static decision environments." It's worth pointing out that the decision environments have to be _very_ simple. Even adding one more mean induces a non-constant (and typically collapsing) bound.

22. Equation above line 405: why repeat that equation, and not repeat Equation. 4? Just curious.

23. Lines 409-11: Couldn't parse.

24. After line 411, we find out that R1+R2=11. This is important and simple; you should tell us in the main text.

25. After line 411: we couldn't parse "allowing us to find the exact tuning of the normative model."

26. In fact, we're lost in pretty much everything between line 411 and the tokens task.

27. Line 429: what's "post-decision token acceleration"?

28. Line 433: "We used three models to fit the subject response data …". As far as we could tell, the three models are continuous time models. How were they adapted to this task, which runs in discrete time? Is it just a matter of making the time step larger?

29. Lines 432-434: please be more clear about parameter counting -- by listing parameters.

30. Lines 437-8: "For more details as to our specific implementation of MCMC for this data, see the MATLAB code available at https://github.com/nwbarendregt/AdaptNormThresh".

We shouldn't have to look at code to get details; all important details should be in the paper.

31. Figure 2—figure supplement 2 and Figure 3—figure supplement 1: we thought the reward changed only once. But it's changing a lot in panel A. What's going on?

32. The Abstract / Introduction isn't clear enough about what you refer to as a "changing / dynamic environment". In particular, there is a rich history of research on environments whose state changes across decisions rather than within individual decisions. Making this distinction explicit, and clarifying that you care about the latter rather than the former should make Abstract / Intro clearer.

33. In the text around Equation. (2), you should mention that you're assuming independence across time.

34. Equation. (3): should c(dt) really be c(t)dt? Its dependence on only the time step size seems incompatible with its initial definition in line 77, where it depends on time t since trial onset. Although eventually, it does become a constant.

35. Below Equation. (3): "We choose generating distributions f_+/- that allow us to explicitly compute the average future value […]" – can you compute the average future value explicitly, or just f_p(p_n+1 | p_n)? Methods only discuss the latter.

36. Figures 2 and 3: the assumed reward/cost magnitudes should be mentioned in the main text, and also if the results were qualitatively dependent on these magnitudes (we assume not?).

37. Figure 2B: "belief" in Bayesian statistics usually refers to a posterior probability, whereas you seem to be using it to refer to log-posterior odds (or log-odds). Please clarify in the text what you mean by "belief" (if you haven't done so already and we missed it). This also refers to Figure 3B and clarifies what the thresholds are on in Figures3/4/5.

38. Figures2C/3C: the letter placements are slightly unclear. In particular, in Figure 2C it is hard to see where exactly 'iv' is placed. Maybe using labeled dots instead would increase placement precision?

39. Line 130: "[…] in which reward fluctuations are governed by a two-state Markov process […]". We couldn't figure out from the description in the main text what setup you are referring to and how to interpret Figure 2 – suppl 3. Please provide more detail (not just in Methods) on the reward switching process: what information is provided to the decision-maker to infer its state, etc.

40. below Line 156: we got lost in the notation for the different noisy / noise-less accumulator models. y_tilde appears to be accumulation with added sensory noise but is in the second point referred to as the "belief y_tilde [of the] normative model", which, being normative, presumably wouldn't have sensory noise. Furthermore, the UGM model seems to use the "noise-free observer's belief y". Is that the belief as defined in Equation. (2) which still includes the sample noise, such that calling it "noise-free" might be confusing?

41. Starting on line 169: the text is unclear on how the models are tuned to cope with the noise, if at all. How the model parameters of the Const and UGM are chosen should also be mentioned in the main text, not just Methods – in particular, that they are tuned to maximize decision performance.

42. Line 332: "+- theta" – missing "(t)"?

43. Line 333: "where observations every dt time units" – fragment?

44. Equation. (10): shouldn't V+ / V- / Vw also be functions of rho?

45. The equation above Equation. (12): how is the expected future value computed? I assume that this can only be done numerically? Either way, please specify the details of how you do so. Referring to a Github repo isn't sufficient.

45. The evidence setup that leads to Equation. (13) appears to be equivalent to the one leading to Equation. (16) in Drugowitsch et al. (2012) for M=1. Is this correct? If yes, is the result equivalent? Either way, the relationship would be worth pointing out.

46. Line 411: "the measured response time T_tilde was drawn from a normal distribution […]" – what happened for predicted response times <0? Did you truncate the normal distribution at 0?

47. Line 432: what was the objective function for the MCMC fits? The joint likelihood of RTs and choices?

48. One of the more realistic scenarios is presented in Figure 2—figure supplement 3, where reward doesn't switch at a fixed time, but uses instead a Markov process. But you do not provide enough details of the task or the results. Is m_R = R_H / R_L? Is it the dark line that corresponds to m_R=\inf (as indicated by legend) or the dotted line (as indicated by caption)? For what value of drift are these thresholds derived? These details should be included.

---

## [Author Response]

I. The tokens task can be analyzed using the formalism introduced in Equation. (3), but it seems pretty far from the "dynamic context" examples emphasized in the bulk of the paper. That doesn't mean the tokens task shouldn't be included. But it does mean we have no evidence one way or the other whether subjects would adopt the highly idiosyncratic boundaries found in simulations (for instance, the infinite threshold boundaries in Figures 2i, 2ii).You need to be clear about this. The way the paper reads, it sounds like you have provided evidence for the dynamic context setup, when in fact that's not the case. It should be crystal clear that dynamic context problems have not been explored, at least not in the lab. Instead, what you showed is that a normative model can beat a particular heuristic model.

We have revised the text substantially to clarify and expand upon these important points. Specifically, we:

a. More clearly define the broad set of possible “dynamic context” conditions, including changes in outcome expectations or evidence quality while the evidence is being processed, where the changes can be either: (1) abrupt, as in the reward-change and SNR-change tasks we introduce, which we analyze only theoretically, or (2) gradual, as in the evidence quality changes in the tokens task, which we analyze theoretically and experimentally (e.g., in Results: Even for such simple tasks, there is a broad set of possible dynamic contexts. In the next section, we will analyze a task where context changes gradually (the tokens task)). Here we focus on tasks where the context changes abruptly.

b. Explain that our theoretical framework is general enough to account for both abrupt and gradual changes clarify that our analysis of data from the tokens task shows that the behavior of subjects is better described by a noisy normative model than by previously considered alternatives applied to that particular form of a dynamic-context task. We also state explicitly that more work is needed to determine if and how people follow normative principles for other dynamic-context tasks, …

II. The following are technical but important.1. Adding noise: you are currently optimizing the model parameters/policy before adding sensory and motor noise. Decision-makers could be aware of sensory noise and so could try to optimize their decision processes with that knowledge. Would you be able to also compare the models' performances if they have been optimized to maximize performance in the presence of all noise? From our understanding, this should be feasible for the Const and UGM model, but might be harder for the normative model. Sensory noise could be included by finding Equation. (13) that includes such noise, but finding the optimal thresholds once RT noise is included might be prohibitive. This is just a suggestion, not an essential inclusion. However, it might be worth at least discussing the difference between what you do, and being clear on the scenario you considered.

We appreciate these important points and now consider them in the revised Discussion. However, we have chosen not to extend our analyses, for several reasons: (1) An optimal observer without internal sensory and motor noise gives the best possible responses, and thus provides a useful benchmark; and (2) we fear that adding results that define optimality with respect to internal sensory and motor noise would, because of the assumptions we would have to make about both the nature and knowledge of those noise sources, be distracting as well as much more speculative and thus make the paper harder to follow.

We have updated the Methods section to highlight these points:

“We have chosen to add these two sources of noise after optimizing each model to maximize average reward rate, rather than reoptimizing each model after adding these additional noise sources. Although we could have reoptimized each model to maximize performance across noise realizations, we were interested in how the models responded to perturbations that drove their performance to be sub-optimal (but possibly near-optimal).”

as well as the Discussion:

“Task-relevant variability can also arise from internal sources, including noise in neural processing of sensory input and motor output (Ma and Jazayeri, 2014; Faisal et al., 2008). We assumed subjects do not have precise knowledge of the strength or nature of these noise sources, and thus they could not optimize their strategy accordingly. However, people may be capable of rapidly estimating performance error that results from such internal noise processes and adjusting on-line (Bonnen et al., 2015). To extend the models we considered, we could therefore assume that subjects can estimate the magnitude of such sensory and motor noise, and use this information to adapt their decision strategies to improve performance.”

2. Fitting token task data: according to Cisek et al. (2009), the same participants performed both the slow and the fast version of the task. However, their fitted reward magnitudes differ by an order of magnitude between the two conditions (your Figure 6C/F). Is it just that the fitting objective didn't well-constrain these parameters? Given that you use MCMC for model fits, you could compare the parameter posteriors across conditions. Furthermore, how much worse would the model fits become if you would fit both conditions simultaneously and share all parameters that can be meaningfully shared across conditions? In any case, an explanation for this difference should be provided in the manuscript.

We now include a supplementary figure (Figure 6—figure supplement 2) comparing the posteriors across conditions as well as reward magnitudes in the slow and fast versions of the tokens task for a representative subject. The maximum likelihood estimate of the reward magnitude tended to be much higher in the slow task than in the fast task. It appears that subjects thus use distinct strategies in the two contexts, which we do not find surprising. We therefore do not expect to obtain fits of the same quality if we assume that subjective reward magnitude is the same across conditions. We speculate that subjects may value reward more in the slow task because it is obtained less frequently. Related effects have been attributed to amplified dopamine responses when rewards are rare (Rothenhoefer et al. 2021 Nat Neurosci). We added text to the Results section to point out this interesting finding:

“This result also shows that, assuming subjects used a normative model, they used distinct model parameters, and thus different strategies, for both the fast and slow task conditions. This finding is clearer when looking at the posterior parameter distribution for each subject and model parameter (see Figure 6—figure supplement 1 for an example). We speculate that the higher estimated value of reward in the slow task may arise due to subjects valuing frequent rewards more favorably.”

3. The dynamic context examples would seem to apply only when subjects take many seconds to make a decision. This would seem to rule out perceptual decision-making tasks. Is this true? If so, you should be upfront about this – so that those who work on perceptual decision-making will know what they're getting into.

We disagree. The impact of normative decision rules is relevant even on shorter timescales, including those relevant to perceptual decisions (e.g., on the order of 100 ms). Figure 2 —figure supplement 2 and Figure 3 —figure supplement 1 demonstrate that even though normative decision rules may invoke plans across multiple context changepoints, often decisions are made within the 1st or 2nd changepoint, and the corresponding reaction time distributions would have a character distinct from those emerging from strategies with flat decision thresholds. Moreover, there is ample evidence that subjects are capable of adapting perceptual evidence integration to sub-second timescales (Ossmy et al. 2013; Glaze et al. 2015). We thus speculate that perceptual decision rules could adapt on similar timescales as predicted by our normative models.

We have updated the Discussion to clarify these points:

“Perceptual decision-making tasks provide a readily accessible route for validating this theory, especially considering the ease with which task difficulty can be parameterized to identify parameter ranges in which strategies can best be differentiated (Philiastides et al. 2006). There is ample evidence already that people can tune the timescale of leaky evidence accumulation processes to the switching rate of an unpredictably changing state governing the statistics of a visual stimulus, to efficiently integrate observations and make a decision about the state (Ossmy et al. 2013; Glaze et al. 2015). We thus speculate that adaptive decision rules could be identified similarly in the strategies people use to make decisions about perceptual stimuli in dynamic contexts.”

4. A known and predictable change in the middle of a task seems somewhat unrealistic. Given that it plays such a central role, concrete examples where this comes up would be very helpful. Or at least you should make a proposal for laboratory experiments where it could come up. The examples in the introduction ("Some of these factors can change quickly and affect our deliberations in real time; e.g., an unexpected shower will send us hurrying down the faster route (Figure 1A), whereas spotting a new ice cream store can make the longer route more attractive.") don't quite fall into the "known and predictable change" category.

Foraging animals must often deal with unpredictable changes in light and visibility conditions, but they also adjust to predictable changes in light brought about by the variation in sunlight with time of day. Sunrise and sunset represent stereotyped changes in foraging conditions as well as necessary escape conditions for prey animals. On shorter timescales, birds and other animals seeking mates, parents, or offspring must often discriminate between two or more calls with known amplitude modulations over time. Financial traders make decisions in markets with fixed open and closing times that strongly shape trading context. Dutch auctions are structured so that an item’s cost is successively lowered until a bidder agrees to pay that amount, reflecting a predictable stair-stepping procedure for cost changes. In all these examples the quality of evidence changes in a predictable way, while the evidence remains noisy.

Concerning laboratory experiments, the first half of the paper already proposes a visual decision-making task. The experiment we analyzed could be implemented as a switching context random dot motion discrimination task with either changes in signal-to-noise (coherence) levels, or changes in reward amounts. Such changes could be signaled or consistently implemented at the same time each trial, so as to be known.

We now have added a sentence in the Introduction:

“People and other animals thus must cope with unpredictable changes in context, such as breaks in the weather (Grubb, 1975), as well as predictable changes that affect their observations, like the daily sunrise and sunset (McNamara et al., 1994).”

as well as a note in the Discussion to indicate the relevance of such task structures, and describe how they can be implemented in a laboratory setting:

“Model-driven experimental design can aid in identification of adaptive decision rules in practice. People commonly encounter unpredictable (e.g., an abrupt thunderstorm) and predictable (e.g., sunset) context changes when making decisions. Natural extensions of common perceptual decision tasks (e.g., random-dot motion discrimination Gold and Shadlen 2002) could include within-trial changes in stimulus signal-to-noise ratio (evidence quality) or anticipated reward payout.”

III. Better contact with existing literature needs to be made. For instance:1. Drugowitsch, Moreno-Bote and Pouget (2014) already computed normative decision policies for time-varying SNR, with the difference that they assumed the SNR to follow a stochastic process rather than a known, deterministic time course. Thus, the work is closely related, but not equivalent.

Indeed we had not explained in detail the differences between their work and ours. We have now added the following sentence to the Discussion to make this clear:

“These strategies include dynamically changing decision thresholds when signal-to-noise ratios of evidence streams vary according to a Cox-Ingersoll-Ross process (Drugowitsch et al., 2014a)”

2. Some early models to predict dynamic decision boundaries were proposed by Busemeyer and Rapoport (1988) and Rapoport and Burkheimer (1971) in the context of a deferred decision-making task.

Thanks very much for pointing out these seminal references, which we now include in the Discussion:

“Several early normative theories were, like ours, based on dynamic programming (Rapoport and Burkheimer, 1971; Busemeyer and Rapoport, 1988), and in some cases models fit to experimental data (Ditterich, 2006).”

3. One of the earliest models to use dynamic programming to predict non-constant decision boundaries was Frazier and Yu (2007). Indeed some boundaries predicted by the authors (e.g. Figure 2v) are very similar to boundaries predicted by this model. In fact, the switch from high to low reward used to propose boundaries in Figure 2v can be seen as a "softer" version of the deadline task in Frazier and Yu (2007).

Again, we very much appreciate the pointer to the very relevant reference, which we include in the Discussion:

“For example, dynamic programming was used to show that certain optimal decisions can require non-constant decision boundaries similar to those of our normative models in dynamic reward tasks (Frazier and Yu, 2007) (Figure 2).”

4. Another early observation that time-varying boundaries can account for empirical data was made by Ditterich (2006). Seems highly relevant to the authors' predictions, but is not cited.

We agree and regret the oversight. We now reference that paper.

5. The authors seem to imply that their results are the first results showing non-monotonic thresholds. This is not true. See, for example, Malhotra et al. (2018). What is novel here is the specific shape of these non-monotonic boundaries.

As with the work by Drugowitsch et al. (2014), this work demonstrates the emergence of non-monotonic boundaries, but in tasks and settings distinct from the ones we consider (which specifically employ dynamic context). We have clarified these points in the manuscript.

IV. Clarity could be massively improved. If you want to write an unclear paper that is your prerogative. However, if you do, you can't say "Our results can aid experimentalists investigating the nuances of complex decision-making in several ways". It would be difficult to aid experimentalists if they have to struggle to understand the paper.Below are comments collected from the three reviewers, and more or less collated (so it's possible there's some overlap, and the order isn't exactly optimized). You can, in fact, almost ignore them, if you take into account the main message: all information should be easily accessible, in the main text, and the figures should be easy to make sense of.As authors, we are aware that the length of replies can sometimes exceed the paper, which is not a good use of anybody's time. Please use your judgment as to which ones you reply to. For instance, if you're going to implement our suggestions, no reason to tell us. Maybe comment only if you have a major objection? Or use some other scheme? What we really care about is that the revised paper is easy to read!

Thanks for providing us with flexibility in how and to what we respond. Generally, we found all comments helpful, and so we have endeavored to make edits that address everything the reviewers brought to our attention. To simplify this letter, we include below only those points that require additional explanation. Otherwise all changes can be found in red in the revised manuscript.

6. Line 76: c(t) is barely motivated at all here. It's better motivated in Methods, but its value is very hard to justify. Why not stick with optimizing average reward, for which c=0? And I don't think you ever told us what c(t) was; only that it was constant (although we could have missed its value).

We have added the following motivation of the cost function *c(t)* to the main text:

“The incremental evidence function *c(t)* represents both explicit time costs, such as a price for gathering evidence, and implicit costs, such as the opportunity cost. While there are many forms of this cost function, we will make the simplifying assumption that it is constant, *c(t)=c*. Because more complex cost functions can influence decision threshold dynamics (Drugowitsch et al., 2012), restricting the cost function to a constant ensures that threshold dynamics are governed purely by changes in the (external) task conditions and not the (internal) cost function.”

We also specified the cost function *c(t)* = 1 that we used in Figure 2-4 in the figure captions. We revised the caption of Figure 5 to make it more clear that we are finding decision threshold motifs as a function of the cost function *c*:

“… B: Colormap of normative threshold dynamics for the “slow'' version of the tokens task in reward-evidence cost parameter space (i.e., as a function of *R_c_* and *c(t) = c* from Equation 3, with punishment *R_i_* set to -1). Distinct …”

We also added in more clarification to the caption of Figure 6C,F to emphasize that we are fitting the cost function *c(t) = c*.

10. Lines 207-9: "Because the total number of tokens was finite and known to the subject, token movements varied in their informativeness within a trial, yielding a dynamic and history-dependent evidence quality that, in principle, could benefit from adaptive decision processes".To us, "history-dependent" implies non-Markov, whereas the tokens task is Markov. But maybe that's not what history-dependent means here? This should be clarified.

Yes, the token count differential is driven by a Markov process, since there is always a 50/50 chance of the token being moved to the top or bottom target. However, the log likelihood ratio associated with either target having more tokens at the end is a non-Markovian, history-dependent process, because the possible LLR increments on each token movement are determined by the token movements so far. This subtlety does make this a dynamic context task, where the evidence quality is the context that changes gradually throughout a trial. We addressed this in our response to the major comments above as we describe the temporal dynamics of the tokens task.

“In addition, the task included two different post-decision token movement speeds, ``slow'' and ``fast'': once the subject committed to a choice, the tokens finished out their animation, moving either once every 170 ms (slow task) or once every 20 ms (fast task). This post-decision movement acceleration changed the value associated with commitment by making the average inter-trial interval (*t_i_* in Equation 1) decrease over time. Because of this modulation, we can interpret the tokens task as a multi-change reward task, where commitment value is controlled through *t_i_* rather than through reward *R_c_*.*”*

19. Equations. 14 and 15 should really be in the main text. They're simple and important.

We added the following text to include these Heaviside functions in the main text and to better motivate our investigation into single-change environments for reward:

“Environments with multiple fluctuations during a single decision lead to complex threshold dynamics, but are comprised of threshold change ``motifs.'' These motifs occur on shorter intervals and tend to emerge from simple monotonic changes in context parameters (Figure 2—figure supplement 2). To better understand the range of possible threshold motifs, we focused on environments with single changes in task parameters. For the reward-change task, we set punishment to *R_i_ = 0*, and assumed reward *R_c_* changes abruptly, so that its dynamics are described by a Heaviside function

Rc(t)=(R1−R2)Hθ(t−0.5)+R1. Thus, the reward switches from a pre-change value of R1 to a post-change value of R2 at *t=0.5*. For this single-change task, …”

and quality:

“In the SNR-change task, optimal strategies for environments with multiple fluctuations are characterized by threshold dynamics adapted to changes in evidence quality in a way similar to changes in reward (Figure 3—figure supplement 1). To study the range of possible threshold motifs, we again considered environments with single changes in evidence quality m=2μ2σ2 by taking *μ* to be a Heaviside function: μ(t)=(μ1−μ2)Hθ(t−0.5)+μ1, For this single-change task, we again found similar threshold motifs to those in the reward-change task (Figure 3A,B).”

23. Lines 409-11: Couldn't parse.

We have revised this paragraph for clarity and to include more details and motivation:

“In addition to the inference noise with strength σy, we also filtered each process through a Gaussian response-time filter with zero mean and standard deviation σmn. Under this response-time filter, if the model predicted a response time *T*, the measured response time T~ was drawn from a normal distribution centered at *T* with standard deviation σmn. If the response time T~ was drawn outside of the simulation's time discretization (i.e., if T~ < 0 or T~> Tf5), we redrew T~ until it fell within the discretization. This filter was chosen to represent both ``early responses'' caused by attentional lapses, as well as ``late responses'' caused by motor processing delays between the formation of a choice in the brain and the physical response.”